# A theory of learning with constrained weight-distribution

**Weishun Zhong**
Harvard and MIT
wszhong@mit.edu

**Ben Sorscher**
Stanford University
bsorsch@stanford.edu

**Daniel D Lee**
Cornell Tech
ddl46@cornell.edu

**Haim Sompolinsky**
Harvard and Hebrew University
hsompolinsky@mcb.harvard.edu

## Abstract

A central question in computational neuroscience is how structure determines function in neural networks. Recent large-scale connectomic studies have started to provide a wealth of structural information such as the distribution of excitatory/inhibitory cell and synapse types as well as the distribution of synaptic weights in the brains of different species. The emerging high-quality large structural datasets raise the question of what general functional principles can be gleaned from them. Motivated by this question, we developed a statistical mechanical theory of learning in neural networks that incorporates structural information as constraints. We derived an analytical solution for the memory capacity of the perceptron, a basic feedforward model of supervised learning, with constraint on the distribution of its weights. Interestingly, the theory predicts that the reduction in capacity due to the constrained weight-distribution is related to the Wasserstein distance between the cumulative distribution function of the constrained weights and that of the standard normal distribution. To test the theoretical predictions, we use optimal transport theory and information geometry to develop an SGD-based algorithm to find weights that simultaneously learn the input-output task and satisfy the distribution constraint. We show that training in our algorithm can be interpreted as geodesic flows in the Wasserstein space of probability distributions. We further developed a statistical mechanical theory for teacher-student perceptron rule learning and ask for the best way for the student to incorporate prior knowledge of the rule (i.e., the teacher). Our theory shows that it is beneficial for the learner to adopt different prior weight distributions during learning, and shows that distribution-constrained learning outperforms unconstrained and sign-constrained learning. Our theory and algorithm provide novel strategies for incorporating prior knowledge about weights into learning, and reveal a powerful connection between structure and function in neural networks.

## 1 Introduction

Learning and memory are thought to take place at the microscopic level by modifications of synaptic connections. Unlike learning in artificial neural networks, synaptic plasticity in the brain operates under structural biological constraints. Theoretical efforts to incorporate some of these constraints have focused largely on the degree of connectivity [17, 35] and the constraints on the sign of the synapses (Excitatory vs. Inhibitory) [4, 16], but few include additional features of synaptic weight distributions observed in the brain [11]. More generally, recent large-scale connectomic studies

36th Conference on Neural Information Processing Systems (NeurIPS 2022).

[36, 59, 62] are beginning to provide a wealth of structural information of neuronal circuits at an unprecedented scope and level of precision, which presents a remarkable opportunity for a more refined theoretical study of learning and memory that takes into account these hitherto unavailable structural information.

Perceptron [56] is arguably the simplest model of computation by single neuron and is the fundamental building block for many modern neural networks. Despite the drastic oversimplification, studying the computational properties of (binary and analog) perceptron has been used extensively in computational neuroscience since its dawn, particularly in the cerebellum (as a model of sensory-motor association) but also in cerebral cortex (for generic associative memory functions) [43, 2, 16, 18, 15, 14]. Forming associations is considered an 'atomic' building block for generic cortical functions, and perceptron memory capacity sets a tight bound on the memory capacity in recurrently connected neuronal circuits with application to cortex and hippocampus [27, 55, 57]. Statistical mechanical analysis predicts that near capacity, an unconstrained perceptron classifying random input-output associations has normally distributed weights [29, 28, 21], see Fig.1(a). In contrast, physiological experiments suggest that biological synapses do not change their excitatory/inhibitory identity during learning (but see recent [33]). In order to take perceptron a step closer to biological realism, prior work has imposed sign constraints during learning [4, 16]. In this case, the predicted weight distribution is a delta-function centered at zero plus a half-normal distribution, see Fig.1(b). However, a wide range of connectomic studies ranging from cortical circuits in animals [38, 32, 47, 74, 62, 40, 9], to human cerebral cortex [47, 62] have shown evidence of lognormally distributed synaptic connections. As an example, Fig.1(c) shows the weight connection distribution in mouse primary auditory cortex (data adapted from [38]). Possible reasons for the ubiquitous lognormal distributions range from biological structural/developmental constraints to computational benefits [64, 60]. Various potential mechanisms for lognormal distributions have been proposed, from multiplicative gradient updates in feedforward and recurrent networks[34, 40, 60], to mixture of additive and multiplicative plasticity rules in spiking networks[30], but the majority of these proposals lead not just to lognormal distributions but also to sparsification in the weights. Instead of adding yet another explanation to the computational origin of lognormal distribution, here we take the observed weight distribution as a prior on the network structure, and ask for its computational consequences. The goal of the paper is to present for the first time a quantitative and qualitative theory of neural network learning performance under non-Gaussian and general weight distributions (not limited to lognormal distributions).

In this paper, we combine two powerful tools: statistical mechanics and optimal transport theory, and present a theory of perceptron learning that incorporates the knowledge of both distribution and sign information as constraints, and gives accurate predictions for capacity and generalization error. Interestingly, the theory predicts that the reduction in capacity due to the constrained weight-distribution is related to the Wasserstein distance between the cumulative distribution function of the constrained weights and that of the standard normal distribution. Along with the theoretical framework, we also present a learning algorithm derived from information geometry that is capable of efficiently finding viable perceptron weights that satisfy desired distribution and sign constraints. This paper is organized as follows: in Section 2.1 we derive the perceptron capacity for classifying random input-output associations using statistical mechanics, and illustrate our theory with a simple example. In Section 3, we derive our learning algorithm using optimal transport theory, and show that distribution of weights found by the learning algorithm coincide with geodesic distributions on a Wasserstein statistical manifold, and therefore training can be interpreted as a geodesic flow. In Section 4 we analyze a parameterized family of biologically realistic weight distributions, and use our theory to predict the shape of the distribution with optimal parameters. We map out the experimental parameter landscape for the estimated distribution of synaptic weights in mammalian cortex and show that our theory's prediction for optimal distribution is close to the experimentally measured value. In Section 5 we further develop a statistical mechanical theory for teacher-student perceptron rule learning and ask for the best way for the student to incorporate prior knowledge about the weight distribution of the rule (i.e., the teacher). Our theory shows that it is beneficial for the learner to adopt different prior weight distributions during learning.

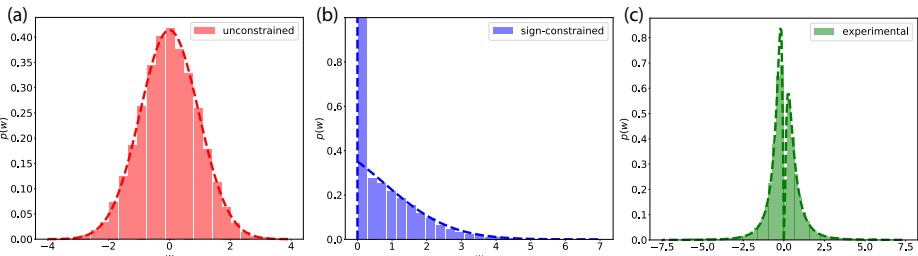

Figure 1: Theoretical and empirical synaptic weight distributions. (a)-(b) predicted distribution following perceptron learning at capacity. (a) Normal distribution when learning is unconstrained. (b) A delta-function plus a half-normal distribution when learning is sign-constrained. (c) Experimentally measured synaptic weight distribution (mouse primary auditory cortex [38]).

## 2 Capacity

### 2.1 Learning under weight distribution constraints

We begin by considering a canonical learning problem: classifying random input-output associations by a perceptron. In biological memory systems, the heavily correlated sensory data is undergoing heavy preprocessing including massive decorrelations, and previous work on brain related perceptron modeling [27, 16, 57] assumes similarly unstructured data. The data consists of pairs $\{\boldsymbol{\xi}^\mu, \zeta^\mu\}_{\mu=1}^P$, where $\boldsymbol{\xi}^\mu$ is an $N$-dimensional random vector drawn i.i.d. from a standard normal distribution, $p(\xi_i^\mu) = \mathcal{N}(0, 1)$, and $\zeta^\mu$ are random binary class labels with $p(\zeta^\mu) = \frac{1}{2}\delta(\zeta^\mu + 1) + \frac{1}{2}\delta(\zeta^\mu - 1)$. The goal is to find a hyperplane through the origin, described by a perceptron weight vector $\boldsymbol{w} \in \mathbb{R}^N$, normalized to $||\boldsymbol{w}||^2 = N$.

We call $\boldsymbol{w}$ a separating hyperplane when it correctly classifies all the examples with margin $\kappa > 0$:

$$\zeta^\mu \frac{\boldsymbol{w} \cdot \boldsymbol{\xi}^\mu}{||\boldsymbol{w}||} \geq \kappa. \tag{1}$$

We are interested in solutions $\boldsymbol{w}$ to Eqn.1 that obey a prescribed distribution constraint, $w_i \sim q(w)$, where $q$ is an arbitrary probability density function. We further demand that $\langle w^2 \rangle_{q(w)} = 1$ to fix the overall scale of the distribution (since the task is invariant to the overall scale of $w$). Thus, the goal of learning is to find weights that satisfy 1 with the additional constraint that the empirical density function $\hat{q}(w) = \frac{1}{N} \sum_i^N \delta(w - w_i)$, formed by the learned weights is similar to $q(w)$, and more precisely that it converges to $q(w)$ as $N \to \infty$ (see Section 2.2 below).

Extension of this setup that includes an arbitrary number of populations each satisfying its own prescribed distribution constraints is discussed in Section 4 and in Appendix A.1.2. Note that the sign constraint is a special case of this scenario with two synaptic populations: one excitatory and one inhibitory. We further discuss the generalization of this setup to include biased inputs and sparse labels in Appendix A.1.3.

### 2.2 Statistical mechanical theory of capacity

We are interested in the thermodynamic limit where $P, N \to \infty$, but the load $\alpha = \frac{P}{N}$ stays $\mathcal{O}(1)$. This limit is amenable to mean-field analysis using statistical mechanics.

Following Gardner's seminal work [29, 28], we consider the fraction $V$ of viable weights that satisfies both Eqn.1 and the distribution constraint $\hat{q} = q$, to all possible weights:

$$V = \frac{\int d\boldsymbol{w} \left[ \prod_{\mu=1}^P \Theta \left( \zeta^\mu \frac{\boldsymbol{w} \cdot \boldsymbol{\xi}^\mu}{||\boldsymbol{w}||} - \kappa \right) \right] \delta(||\boldsymbol{w}||^2 - N) \delta \left( \int dk \, (\hat{q}(k) - q(k)) \right)}{\int d\boldsymbol{w} \delta(||\boldsymbol{w}||^2 - N)}. \tag{2}$$

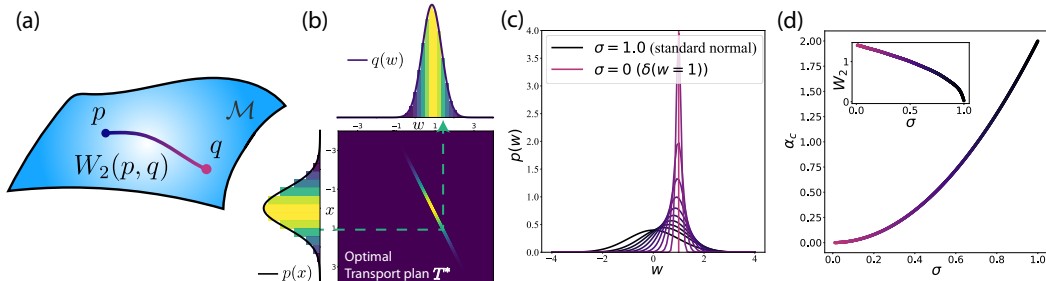

Figure 2: An illustration of optimal transport from a standard normal distribution $\mathcal{N}(0,1)$ to normal distributions with nonzero mean $\mathcal{N}(\sqrt{1-\sigma^2}, \sigma^2)$. (a) A schematic of the space $(\mathcal{M}, W_2)$ of probability distributions. (b) An example optimal transport plan from standard normal, $p(x)$, to a normal with $\sigma = 0.5$, $q(w)$. The optimal transport plan $T^*$ is plotted in between the distributions. $T^*$ moves $p(x)$ units of probability mass $x$ to location $w$, as indicated by the dashed line, and the colors are chosen to reflect the amount of probability mass to be transported. (c) $\mathcal{N}(\sqrt{1-\sigma^2}, \sigma^2)$ interpolates between standard normal ($\sigma = 0$) to a $\delta$-function at 1 ($\sigma = 1$). (d) Capacity $\alpha_c(\kappa = 0)$ as a function of $\sigma$. Inset shows the $W_2$ distance as a function of $\sigma$.

In Eqn.2, we impose the distribution constraint $\hat{q} = q$ by demanding that in the thermodynamic limit, all Fourier modes of $q$ and $\hat{q}$ are the same , i.e., that $q(k) = \int dw e^{ikw} q(w) = \hat{q}(k) = \frac{1}{N} \sum_i^N e^{ikw_i}$, where in the last equality we have used the definition of empirical distribution. We perform a quenched average over random patterns $\boldsymbol{\xi}^\mu$ and labels $\zeta^\mu$. This amounts to calculating $\langle \log V \rangle$, which can be done using the replica trick [29, 28].

We focus on solutions with maximum margin $\kappa$ at a given load $\alpha$, or equivalently, the maximum load capacity $\alpha_c(\kappa)$ of separable patterns given margin $\kappa$. We proceed by assuming replica symmetry in our mean field analysis, which in general might not hold because the constraint $\hat{q} = q$ is non-convex. For all the results presented in the main text, replica symmetry solution is supported by numerical simulations. In Appendix A.5 we explore the validity of replica symmetric solutions in the case of strongly bimodal distributions and show that they fail only very close to the binary (Ising) limit.

Detailed calculations of the mean-field theory are presented in Appendix A.1.1. Our mean-field theory predicts that the reduction in capacity due to the distribution constraint is proportional to the Jacobian of the transformation from $w \sim q(w)$ to a normally distributed variable $x(w) \sim \mathcal{N}(0,1)$,

$$\alpha_c(\kappa) = \alpha_0(\kappa) \left\langle \frac{dw}{dx} \right\rangle_x^2, \tag{3}$$

where $\alpha_0(\kappa) = \left[ \int_{-\kappa}^\infty Dt(\kappa+t)^2 \right]^{-1}$ is the capacity of an unconstrained perceptron, from Gardner theory [29, 28], and $\kappa = 0$ reduces to the classical result of $\alpha_0(0) = 2$. The Jacobian factor, $\langle dw/dx \rangle_x$, can be written in terms of the constrained distribution's cumulative distribution function (CDF), $Q(w)$, and the standard normal CDF $P(x) = \frac{1}{2} \left[ 1 + \mathrm{Erf}(\frac{x}{\sqrt{2}}) \right]$, namely,

$$\left\langle \frac{dw}{dx} \right\rangle_x = \int_0^1 du Q^{-1}(u) P^{-1}(u). \tag{4}$$

Note that since the second moments are fixed to unity, $0 \leq \left\langle \frac{dw}{dx} \right\rangle_x \leq 1$ and it equals 1 iff $p = q$.

## 2.3 Geometrical interpretation of capacity

The jacobian factor Eqn.4 can be rewritten as

$$\left\langle \frac{dw}{dx} \right\rangle_x = 1 - \frac{1}{2} W_2(Q, P)^2, \tag{5}$$

where $W_k$ ($k = 2$ in above) is the Wasserstein-$k$ distance, given by

$$W_k(Q, P) = \left[ \int_0^1 du \left( Q^{-1}(u) - P^{-1}(u) \right)^k \right]^{1/k}. \tag{6}$$

[In the following, we will make frequent use of both the probability density function (PDF), and the cumulative distribution function (CDF). We distinguish them by using upper case letters for CDFs, and lower case letters for PDFs.]

The Wasserstein distance measures the dissimilarity between two probability distributions, and is the geodesic distance between points on the manifold of probability distributions [41, 25, 20]. Therefore, we can interpret Eqn.3 as predicting that the reduction in memory capacity tracks the geodesic distance we need travel from the standard normal distribution $P$ to the target distribution $Q$ (Fig.2(a)).

We demonstrate Eqn.3 and Eqn.5 with an instructive example. Let's consider a parameterized family of normal distributions, with the second moment fixed to 1: $q(w) = \mathcal{N}(\sqrt{1 - \sigma^2}, \sigma^2)$, see Fig.2(c). At $\sigma = 1$, $q(w)$ is the standard normal distribution and we recover the unconstrained Gardner capacity $\alpha_0(\kappa = 0) = 2$. As $\sigma \to 0$, $q(w)$ becomes a $\delta$-function at 1 and $\alpha_c(\kappa) \to 0$ (Fig.2(c)).

As evident in this simple example, perceptron capacity is strongly affected by its weight distribution. Our theory enables prediction of the shape of the distribution with optimal parameters within a parameterized family of distributions. We apply our theory to a family of biologically plausible distributions and compare our prediction with experimentally measured distributions in Section 4.

## 3 Optimal transport and the DisCo-SGD learning algorithm

Eqn.3 predicts the storage capacity for a perceptron with a given weight distribution, but it does not specify a learning algorithm for finding a solution to this non-convex learning problem. Here we present a learning algorithm for perceptron learning with a given weight distribution constraint. This algorithm will also serve to test our theoretical predictions. For this purpose, we use optimal transport theory to develop an SGD-based algorithm that is able to find max-margin solutions that obey the prescribed distribution constraint. Furthermore, we show that training can be interpreted as traveling along the geodesic connecting the current empirical distribution and the target distribution.

Stochastic gradient descent (SGD) on a cross-entropy loss has been shown to asymptotically converge to max-margin solutions on separable data [63, 50]. Given data $\{\boldsymbol{\xi}^\mu, \zeta^\mu\}_{\mu=1}^P$, we use logistic regression to predict class labels from our perceptron weights, $\hat{\zeta}^\mu = \sigma(\boldsymbol{w}^t \cdot \boldsymbol{\xi}^\mu)$, where $\sigma(z) = (1 + e^{-z})^{-1}$ and $\boldsymbol{w}^t$ is the weight at the $t$-th update. This defines an SGD update rule :

$$w_i^{t+\delta t} \leftarrow w_i^t - \delta t \sum_\mu \xi_i^\mu (\hat{\zeta}^\mu - \zeta^\mu), \tag{7}$$

where the $\mu$-summation goes from 1 to $P$ for full-batch GD and goes from 1 to mini-batch size $B$ for mini-batches SGD (see Appendix A.4 for more details). The theory of optimal transport provides a principled way of transporting each individual weight $w_i^t$ to a new value so that overall the new set of weights satisfies the prescribed target distribution. In 1-D, the optimal transport plan $T^*$ has a closed-form solution in terms of the current CDF $P$ and target CDF $Q$ [66, 3]: $T^* = Q^{-1} \circ P$, where $\circ$ denotes functional composition. We demonstrate the optimal transport map in Fig.2(b) for the instructive example discussed in Section 2.3.

In order to apply $T^*$ to transport our weights $\{w_i\}$ (omitting superscript $t$), we form the empirical CDF $\hat{Q}(w) = \frac{1}{N} \sum_{i=1}^N \mathbf{1}_{w_i \leq w}$, which counts how many weights $w_i$ are observed below value $w$. Then the new set of weights $\{\hat{w}_i\}$ satisfying target CDF $Q$ can be written as

$$\hat{w}_i = Q^{-1} \circ \hat{Q}(w_i). \tag{8}$$

We illustrate Eqn.8 in action in Table 1(b).

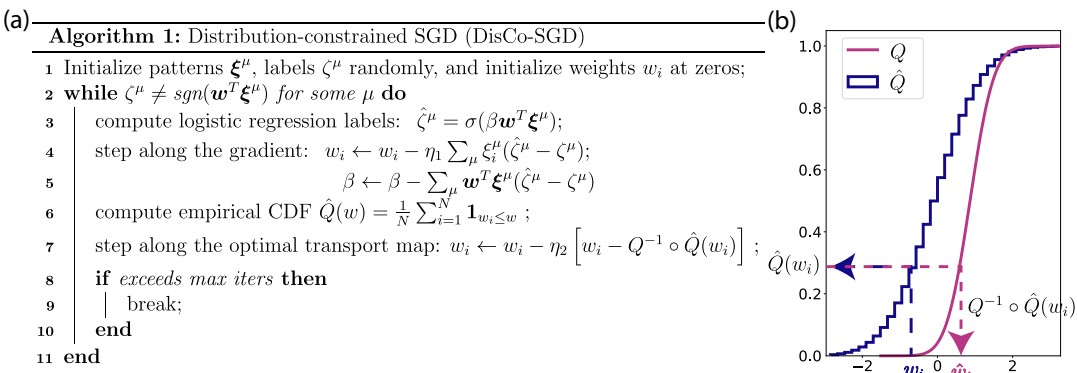

**Algorithm 1:** Distribution-constrained SGD (DisCo-SGD)

1  Initialize patterns $\boldsymbol{\xi}^\mu$, labels $\zeta^\mu$ randomly, and initialize weights $w_i$ at zeros;
2  **while** $\zeta^\mu \neq sgn(\boldsymbol{w}^T\boldsymbol{\xi}^\mu)$ *for some $\mu$* **do**
3      compute logistic regression labels: $\hat{\zeta}^\mu = \sigma(\beta\boldsymbol{w}^T\boldsymbol{\xi}^\mu)$;
4      step along the gradient: $w_i \leftarrow w_i - \eta_1 \sum_\mu \xi_i^\mu(\hat{\zeta}^\mu - \zeta^\mu)$;
5      $\qquad\qquad \beta \leftarrow \beta - \sum_\mu \boldsymbol{w}^T\boldsymbol{\xi}^\mu(\hat{\zeta}^\mu - \zeta^\mu)$
6      compute empirical CDF $\hat{Q}(w) = \frac{1}{N}\sum_{i=1}^N \mathbf{1}_{w_i \leq w}$ ;
7      step along the optimal transport map: $w_i \leftarrow w_i - \eta_2 \left[ w_i - Q^{-1}\circ\hat{Q}(w_i) \right]$ ;
8      **if** *exceeds max iters* **then**
9          break;
10     **end**
11 **end**

Table 1: Disco-SGD algorithm. (a) We perform alternating steps of gradient descent along the cross-entropy loss (Eqn.7), followed by steps along the optimal transport direction (Eqn.9). (b) An illustration of Eqn.8. For a given $w_i$, we first compute its empirical CDF value $\hat{Q}(w_i)$, then use the inverse target CDF to transport $w_i$ to its new value, $\hat{w}_i = Q^{-1}\left(\hat{Q}(w_i)\right)$.

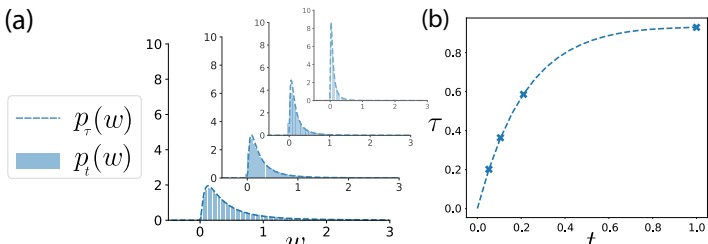

Figure 3: Intermediate distributions during learning are on the geodesic. (a) The solid histograms are the intermediate distribution $p_t$ at different training time $t$ from the DisCo-SGD algorithm, the dashed lines are geodesic distributions $p_\tau$ with the same $W_2$ distance to the target distribution $Q$. From right to left the training time advances, and the distributions transform further away from the $\delta$-function initialization, and approach the target distribution (a lognormal, in this example). (b) The geodesic time $\tau$ as a function of the training time $t$. Location of the crosses correspond to the distributions shown in (a).

However, performing such a one-step projection strongly interferes with the cross-entropy objective, and numerically often results in solutions that do not perfectly classify the data. Therefore, it would be beneficial to have an incremental update rule based on Eqn.8:

$$w_i^{\tau+\delta\tau} \leftarrow w_i^\tau + \delta\tau\left(\hat{w}_i - w_i^\tau\right), \tag{9}$$

where we have used a different update time $\tau$ to differentiate with the cross-entropy update time $t$.

We present our complete algorithm in Table 1(a), which we named 'Distribution-constrained SGD' (DisCo-SGD) algorithm. In the DisCo-SGD algorithm, we perform alternating updates on Eqn.7 and Eqn.9, and identify $\delta t$ and $\delta\tau$ as learning rates $\eta_1$ and $\eta_2$. Note that in logistic regression, the norm of the weight vector $||\boldsymbol{w}||$ is known to increase with training and the max-margin solution is only recovered at $||\boldsymbol{w}|| \to \infty$. In contrast, imposing a distribution constraint fixes the norm. Therefore, to allow a variable norm, in Table 1 we include a trainable parameter $\beta$ in our algorithm to serve as the norm of the weight vector. This algorithm allows us to reliably discover linearly separable solutions obeying the prescribed weight distribution $Q$.

Interestingly, Eqn.9 takes a similar form to geodesic flows in Wasserstein space. Given samples $\{w_i\}$ drawn from the initial distribution $P$ and $\{\hat{w}_i\}$ drawn from the final distribution $Q$, samples $\{w_i^\tau\}$ from intermediate distributions $P_\tau$ along the geodesic can be calculated as $w_{(i)}^\tau = (1-\tau)w_{(i)} + \tau\hat{w}_{(i)}$, where subscript $(i)$ denotes ascending order (see more in Appendix A.2). For intermediate perceptron

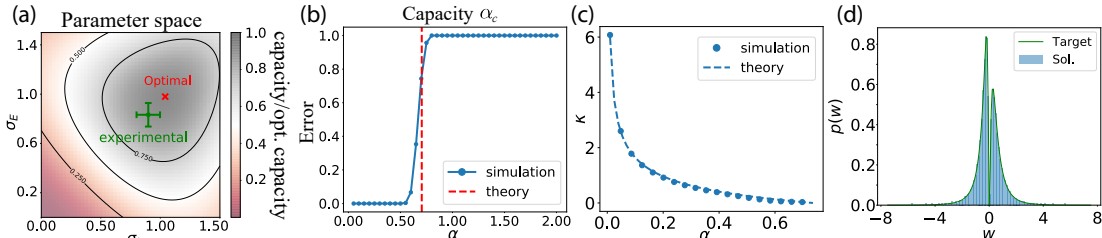

Figure 4: Biologically-realistic distribution and parameter landscape. (a) Capacity (normalized by the optimal value in the landscape) as a function of the lognormal parameters $\sigma_E$ and $\sigma_I$. Experimental value is shown in green with error bars, and optimal capacity is shown in red. (b)-(d) (theory from Eqn.10 and simulations from DisCo-SGD): (b) Determination of capacity; (c) Max-margin $\kappa$ at different load $\alpha$, which is the same as $\alpha_c(\kappa)$; (d) Example weight distribution obtained in simulation.

weights $\boldsymbol{w}^t$ found by our algorithm, we can compute its empirical distribution $p_t$ and compare with theoretical distribution $p_\tau$ along the geodesic with the same $W_2$ distance to the target distribution (see Appendix A.2 for how to calculate $p_\tau$). In Fig.3(a), we show that indeed the empirical distributions $p_t$ agree with the geodesic distributions $p_\tau$ at geodesic time $\tau(t)$ (Fig.3(a)). The relation between the geodesic time $\tau$ and the SGD update time $t$ is shown in Fig.3(b). The interplay between the cross-entropy objective and the distribution constraint is manifested in the rate at which the distribution moves along the geodesic between the initial distribution and the target one.

## 4 Biologically-realistic distribution (E/I balanced lognormals) and experimental landscape

In order to apply our theory to the more biologically-realistic cases, we generalize our theory from a single prescribed distribution to an arbitrary number of input subpopulations each obeys its own distribution. We consider a perceptron that consists of $M$ synaptic populations $\boldsymbol{w}^m$ indexed by $m$, each constrained to satisfy its own weight distribution $w_i^m \sim q_m(w^m)$. We denote the overall weight vector as $\boldsymbol{w} \equiv \{\boldsymbol{w}^m\}_{m=1}^M \in \mathbb{R}^{N \times 1}$, where the total number of weights is $N = \sum_{m=1}^M N_m$. In this case, the capacity Eqn.3 is generalized to (See Appendix A.1.2 for detailed derivation):

$$\alpha_c(\kappa) = \alpha_0(\kappa) \left[ \sum_m^M g_m \left\langle \frac{dw^m}{dx} \right\rangle_x \right]^2, \tag{10}$$

where $g_m = N_m/N$ is the fraction of weights in this population. Eqn. 10 allows us to investigate the parameter space of capacity with biologically-realistic distributions and compare with the experimentally measured values. In particular, we are interested the case with two synaptic populations that models the excitatory/inhibitory synpatic weights of a biological neuron, hence, $m = E, I$. We model the excitatory/inhibitory synaptic weights as drawn from two separate lognormal distributions $(g_I = 1 - g_E)$: $w_i^E \sim \frac{1}{\sqrt{2\pi}\sigma_E w^E} \exp\left\{-\frac{(\ln w^E - \mu_E)^2}{2\sigma_E^2}\right\}$ and $w_i^I \sim \frac{1}{\sqrt{2\pi}\sigma_I w^I} \exp\left\{-\frac{(\ln w^I - \mu_I)^2}{2\sigma_I^2}\right\}$.

We also demand that the mean synaptic weights satisfy the E/I balance condition [70, 71, 68, 69, 57, 48, 18] $g_E \langle w^E \rangle = g_I \langle w^I \rangle$ as is often observed in cortex connectomic experiments [5, 73, 51, 53, 8]. With the E/I balance condition and fixed second moment, the capacity is a function of the lognormal parameters $\sigma_E$ and $\sigma_I$. In Fig.4(a) we map out the 2d parameter space of $\sigma_E$ and $\sigma_I$ using Eqn.10, and find that the optimal choice of parameters which yields the maximum capacity solution is close to the experimentally measured values in a recent connectomic studies in mouse primary auditory cortex [38].

In order to test our theory's validity on this estimated distribution of synaptic weights, we perform DisCo-SGD simulation with model parameters $\sigma_E$ and $\sigma_I$ fixed to their experimentally measured values. Both the capacity (Fig.4(b)), max-margin $\kappa$ at different load (Fig.4(c)), and the empirical weights found by the algorithm (Fig.4(d)) are in good agreement with our theoretical prediction.

# 5 Generalization performance

## 5.1 Distribution-constrained learning as circuit inference

A central question in computational neuroscience is how underlying neural circuits determine its computation. Recently, thanks to new parallelized functional recording technologies, simultaneous recordings of the activity of hundreds of neurons in response to an ensemble of inputs are possible [1, 12]. An interesting challenge is to infer the structural connectivity from the measured input-output activity patterns. It is interesting to ask how are these stimuli-response relations related to the underlying structure of the circuit [54, 39]. In the following, we try to adress this circuit reconstruction task in a simple setup where a student perceptron tries to learn from a teacher perceptron [61, 22]. In this setup, the teacher is considered to be the underlying ground-truth neural circuit. The student is attempting to infer the connection weights of this ground-truth circuit by observing a series of input-output relations generated by the teacher. After learning is completed, one can assess the faithfulness of the inference by comparing the teacher and student. The teacher-student setup is also a well-known 'toy model' for studying generalization performance [42, 37, 44]. In this case since the learning data are generated by the teacher, the overlap between teacher and student determines the generalization performance of the learning. Here we ask to what extent prior knowledge of the teacher weight distribution helps in learning the rule and how this knowledge can be incorporated in learning. A similar motivation may arise in other contexts, in which there is a prior knowledge about the weight distribution of an unknown target linear classifier.

Let's consider the teacher perceptron, $\boldsymbol{w}_t \in \mathbb{R}^N$, drawn from some ground-truth distribution $p_t$. Given random inputs $\boldsymbol{\xi}^\mu$ with $p(\xi_i^\mu) = \mathcal{N}(0, 1)$, we generate labels by $\zeta^\mu = \text{sgn}(\boldsymbol{w}_t \cdot \boldsymbol{\xi}^\mu / ||\boldsymbol{w}_t|| + \eta^\mu)$, where $\eta^\mu$ is input noise and $\eta^\mu \sim \mathcal{N}(0, \sigma^2)$. We task the student perceptron $\boldsymbol{w}_s$ to find the max-margin linear classifier for data $\{\boldsymbol{\xi}^\mu, \zeta^\mu\}_{\mu=1}^p$: $\max \kappa : \zeta^\mu \boldsymbol{w}_s \cdot \boldsymbol{\xi}^\mu \geq \kappa ||\boldsymbol{w}_s||$. Let's define the teacher-student overlap as

$$R = \frac{\boldsymbol{w}_s \cdot \boldsymbol{w}_t}{||\boldsymbol{w}_s|| \, ||\boldsymbol{w}_t||}, \tag{11}$$

which is a measure the faithfulness of the circuit inference. The student's generalization error is then related to the overlap by $\varepsilon_g = 1/\pi \arccos \left( R/\sqrt{1 + \sigma^2} \right)$ [61, 22].

As a baseline, let's first consider a totally uninformed student (without any structural knowledge of the teacher), learning from a teacher with a given (in particular non-Gaussian) weight distribution. In this case, we can determine the overlap $R$ (Eqn.11) as a function of load $\alpha$ by solving the replica symmetric mean field self-consistency equations as in [61, 22]. An example of such learning for a lognormal teacher distribution is shown in Fig.5(a) ('unconstrained') for the noiseless case ($\sigma = 0$). Note that in the presence of noise in the labels ($\sigma \neq 0$), $\alpha$ is bounded by $\alpha_c(\sigma)$, since max-margin learning of separable data is assumed. The case with nonzero $\sigma$ is presented in Appendix A.3.4. In this unconstrained case, the student's weight distribution evolves from a Gaussian for low $\alpha$ to one which increasingly resembles the teacher distribution for large $\alpha$ (Fig.5(b)).

Next, we consider a student with information about the signs of the individual teacher weights. We can apply this knowledge as a constraint and demand that the signs of individual student weights agree with that of the teacher's. The additional sign-constraints require a modification of replica calculation in [61, 22], which we present in Appendix A.3.1. Surprisingly, we find both analytically and numerically that if the teacher weights are not too sparse, the max-margin solution generalizes poorly: after a single step of learning (with random input vectors), the overlap, $R$, drops substantially from its initial value (see 'sign-constrained' in Fig.5(a)). The source of the problem is that, due to the sign constraint, max-margin training with few examples yields a significant mismatch between the student and teacher weight distributions. After only a few steps of learning, half of the student's weights are set to zero, and the student's distribution, $p(w_s) = 1/2\delta(0) + 1/\sqrt{2\pi} \exp\{-w_s^2/4\}$, deviates significantly from the teacher's distribution (see more in Appendix A.3.3). The discrepancy between the teacher and student weight distributions therefore suggest that we should incorporate distribution-constraint into learning.

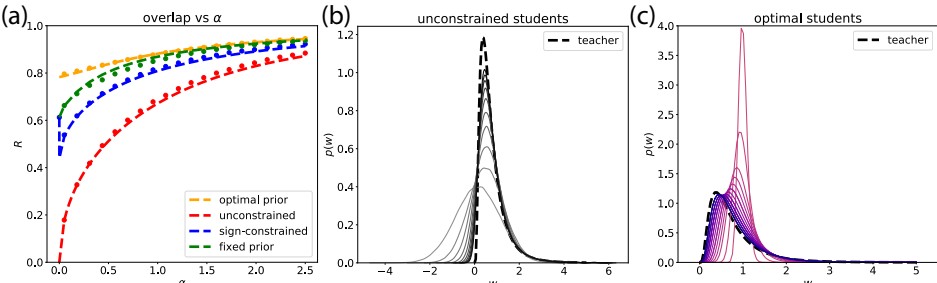

Figure 5: Compare different learning paradigms. (a) Teacher-student overlap $R$, or equivalently the generalization error $\varepsilon_g = 1/\pi \arccos R$, as a function of load $\alpha$ in different learning paradigms. Dashed lines are from theory, and dots are from simulation. Note that there is an initial drop of the overlap in sign-constrained learning due to sparsification of weights. (b)-(c) The darker color curves correspond to larger $\alpha$, and dashed line is teacher distribution (same in both cases). (b) Distribution of an unconstrained student evolves from normal distribution toward the teacher distribution. (c) Optimal student prior evolves from a $\delta$-function toward the teacher distribution.

## 5.2 Distribution-constrained learning outperforms unconstrained and sign-constrained learning

Let's consider the case that the student weight are constrained to some *prior* distribution $q_s(w_s)$, while the teacher obeys a distribution $p_t(w_t)$, for an arbitrary pair $q_s, p_t$. We can write down the Gardner volume $V_g$ for generalization as in the capacity case (Eqn.2):

$$
V_g = \frac{\int d\boldsymbol{w}_s \left[ \prod_{\mu=1}^{P} \Theta \left( \mathrm{sgn}\left( \frac{\boldsymbol{w}_t \cdot \boldsymbol{\xi}^\mu}{||\boldsymbol{w}_t||} + \eta^\mu \right) \frac{\boldsymbol{w}_s \cdot \boldsymbol{\xi}^\mu}{||\boldsymbol{w}_s||} - \kappa \right) \right] \delta(||\boldsymbol{w}_s||^2 - N) \delta\left( \int dk \, (\hat{q}(k) - q(k)) \right)}{\int d\boldsymbol{w}_s \delta(||\boldsymbol{w}_s||^2 - N)}.
$$
(12)

To obtain ensemble average of system over different realizations of the training set, we perform the quenched average of $\log V_g$ over the patterns $\boldsymbol{\xi}^\mu$ and teacher $\boldsymbol{w}_t$, and consider the thermodynamic limit of $N, P \to \infty$ and $\alpha = \frac{P}{N}$ stays $\mathcal{O}(1)$. We use the replica trick similar to [61, 22]. Overlap $R$ (Eqn.11) can be determined as a function of load $\alpha$ by solving the replica symmetric mean field self-consistency equations in Appendix A.3.2. In this distribution-constrained setting, we can perform numerical simulations with DisCo-SGD algorithm (Table 1) to find such weights and compare with the predictions of our theory.

Now we ask if the student has a *prior* on the teacher's weight distribution $p_t$, whether incorporating this knowledge in training will improve generalization performance. One might be tempted to conclude that the optimal prior distribution the student should adopt is always that of the teacher's, i.e., $q_s = p_t$. We call this learning paradigm 'fixed prior', and show that its generalization performance is better than that of the unconstrained and sign-constrained case (Fig.5(a)). However, instead of using a fixed prior for the student, we can in fact choose the *optimal prior* distribution $p_s^*$ at different load $\alpha$. This presents a new learning paradigm we called 'optimal prior'. In Fig.5(a), we show that choosing optimal priors at different $\alpha$ achieves the overall best generalization performance compared with all other learning paradigms. For a given parameterized family of distributions, our theory provides a way to analytically obtain the optimal prior $p_s^*$ as a function of $\alpha$ (Fig.5(c)). Note that unlike the unconstrained case (Fig.5(b)), the optimal prior starts from a $\delta$-function at 1 at zero $\alpha$, and asymptotically approaches the teacher distribution $p_t$ as $\alpha \to \infty$.

## 6 Summary and Discussion

We have developed a statistical mechanical framework that incorporates structural constraints (sign and weight distribution) into perceptron learning. The synaptic weights in our perceptron learning satisfy two key biological constraints: (1) individual synaptic signs are not affected by the learning task (2) overall synaptic weights obey a prescribed distribution. These constraints may arise also in

neuromorphic devices [31, 67]. Under the replica-symmetry assumption, we derived a novel form of distribution-constrained perceptron storage capacity, which admits a simple geometric interpretation of the reduction in capacity in terms of the Wasserstein distance between the standard normal distribution and the imposed distribution. To numerically test our analytic theory, we used tools from optimal transport and information geometry to develop an SGD-based algorithm, DisCo-SGD, in order to reliably find weights that satisfy such prescribed constraints and correctly classify the data, and showed that training with the algorithm can be interpreted as geodesic flows in the Wasserstein space of distributions. It would be interesting to compare our theory and algorithm to [7, 58] where the Wasserstein distance is used as an objective for training generative models. We applied our theory to the biologically realistic case of of excitatory/inhibitory lognormal distributions that are observed in the cortex, and found experimentally-measured parameters close to the optimal parameter values predicted by our theory. We further studied input-output rule learning where the target rule is defined in terms of a weighted sum of the inputs, and asked to what extent prior knowledge of the target distribution may improve generalization performance. Using the teacher-student perceptron learning setup, we showed analytically and numerically that distribution constrained learning substantially enhances the generalization performance. In the context of circuit inference, distribution constrained learning provides a novel and reliable way to recover the underlying circuit structure from observed input-output neural activities. In summary, our work provides new strategies of incorporating knowledge about weight distribution in neural learning and reveals a powerful connection between structure and function in neural networks. Ongoing extensions of the present work include weight distribution constraints in recurrent and deep architectures as well as testing against additional connectomic databases.

## Acknowledgments

This paper is dedicated to the memory of Mrs. Lily Safra, a great supporter of brain research. The authors would like to thank Madhu Advani, Haozhe Shan, and Julia Steinberg for very helpful discussions. W.Z. acknowledges support from the Swartz Program in Theoretical Neuroscience at Harvard and the NIH grant NINDS (1U19NS104653). B.S. acknowledges support from the Stanford Graduate Fellowship. D.D.L. acknowledges support from the NIH grant NINDS (1U19NS104653). H.S. acknowledges support from the Swartz Program in Theoretical Neuroscience at Harvard, the NIH grant NINDS (1U19NS104653), and the Gatsby Charitable Foundation.

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
