# A  Appendix

## Preliminaries

Throughout the appendix, we make frequent use of Gaussian integrals. We introduce short-hand notations $\int Dt \equiv \int \frac{dt}{\sqrt{2\pi}} e^{-t^2/2}$ and $H(x) \equiv \int_x^\infty Dt$. Also, when we do not specify the integration range it is understood that we are integrating from $-\infty$ to $\infty$.

## A.1  Capacity supplemental materials

### A.1.1  Replica calculation of distribution-constrained capacity

In this section, we present the replica calculation of the distribution-constrained storage capacity of a perceptron.

As described in main text Eqn.2, we need to perform a quenched average $\langle \cdot \rangle$ over the patterns $\boldsymbol{\xi}^\mu$ and labels $\zeta^\mu$ for $\log V$, which can be carried out using the replica trick, $\langle \log V \rangle = \lim_{n\to 0}(\langle V^n \rangle - 1)/n$. Following [29, 28], we consider first integer $n$, and at the end perform analytic continuation of $n \to 0$. The replicated Gardner volume is:

$$V = \frac{\prod_{\alpha=1}^n \int d\boldsymbol{w}^\alpha \left[ \prod_{\mu=1}^P \Theta \left( \zeta^\mu \frac{\boldsymbol{w}^\alpha \cdot \boldsymbol{\xi}^\mu}{||\boldsymbol{w}^\alpha||} - \kappa \right) \right] \delta(||\boldsymbol{w}^\alpha||^2 - N) \delta \left( \int dk \left( \hat{q}(k) - q(k) \right) \right)}{\prod_{\alpha=1}^n \int d\boldsymbol{w}^\alpha \delta(||\boldsymbol{w}^\alpha||^2 - N)} \tag{13}$$

Let's rewrite the Heaviside step function using Fourier representation of the $\delta$-function $\delta(x) = \int_{-\infty}^\infty \frac{dk}{2\pi} e^{ikx}$ as (defining $z_\alpha^\mu = \zeta^\mu \frac{\boldsymbol{w}^\alpha \cdot \boldsymbol{\xi}^\mu}{||\boldsymbol{w}^\alpha||}$)

$$\Theta \left( z_\alpha^\mu - \kappa \right) = \int_\kappa^\infty d\rho_\alpha^\mu \delta(\rho_\alpha^\mu - z_\alpha^\mu) = \int_\kappa^\infty d\rho_\alpha^\mu \int \frac{dx_\alpha^\mu}{2\pi} e^{ix_\alpha^\mu(\rho_\alpha^\mu - z_\alpha^\mu)}. \tag{14}$$

Note that now all the $\boldsymbol{\xi}^\mu, \zeta^\mu$ dependence is in $e^{-ix_\alpha^\mu z_\alpha^\mu}$. We perform the average with respect to $\xi_i^\mu \sim p(\xi_i^\mu) = \mathcal{N}(0,1)$ and $p(\zeta^\mu) = \frac{1}{2}\delta(\zeta^\mu + 1) + \frac{1}{2}\delta(\zeta^\mu - 1)$ (also note that $||\boldsymbol{w}^\alpha|| = \sqrt{N}$):

$$\left\langle \prod_{\mu\alpha} e^{-ix_\alpha^\mu z_\alpha^\mu} \right\rangle_{\xi\eta} = \prod_{\mu j} \left\langle \exp \left\{ -\frac{i}{\sqrt{N}} \zeta^\mu \xi_j^\mu \sum_\alpha x_\alpha^\mu w_j^\alpha \right\} \right\rangle_{\xi\zeta}$$

$$= \prod_{\mu i} \left\langle \exp \left\{ -\frac{(\zeta^\mu)^2}{2N} \sum_{\alpha\beta} x_\alpha^\mu x_\beta^\mu w_i^\alpha w_i^\beta \right\} \right\rangle_\zeta \tag{15}$$

$$= \prod_\mu \exp \left\{ -\frac{1}{2N} \sum_{\alpha\beta} x_\alpha^\mu x_\beta^\mu \sum_i w_i^\alpha w_i^\beta \right\}.$$

Introducing the replica overlap parameter $q_{\alpha\beta} = \frac{1}{N} \sum_i w_i^\alpha w_i^\beta$, and notice that the $\mu$ index gives $P$ identical copies of the same integral. We can suppress the $\mu$ indices and write

$$\left\langle \prod_{\mu\alpha} \Theta \left( z_\alpha^\mu - \kappa \right) \right\rangle_{\xi\zeta} = \left[ \int_\kappa^\infty \left( \prod_\alpha \frac{d\rho_\alpha dx_\alpha}{2\pi} \right) e^K \right]^P, \tag{16}$$

where

$$K = i \sum_\alpha x_\alpha \rho_\alpha - \frac{1}{2} \sum_{\alpha\beta} q_{\alpha\beta} x_\alpha x_\beta \tag{17}$$

captures all the data dependence in the quenched free energy landscape, and therefore it is called the 'energetic' part of the free energy. In contrast, the $\delta$-functions in Eqn.13 are called 'entropic' part because they regulate what kind of weights are considered in the version space (space of viable weights).

**The entropic part**

$$\delta(Nq_{\alpha\beta} - \sum_i w_i^\alpha w_i^\beta) = \int \frac{d\hat{q}_{\alpha\beta}}{2\pi} \exp\left\{ iN\hat{q}_{\alpha\beta} q_{\alpha\beta} - i\hat{q}_{\alpha\beta} \sum_i w_i^\alpha w_i^\beta \right\}. \tag{18}$$

Note that the normalization constraint $\delta(||\boldsymbol{w}^\alpha||^2 - N)$ is automatically satisfied by requiring $q_{\alpha\alpha} = 1$. Using replica-symmetric ansatz: $\hat{q}_{\alpha\beta} = -\frac{i}{2}(\Delta\hat{q}\delta_{\alpha\beta} + \hat{q}_1)$, and $q_{\alpha\beta} = (1-q)\delta_{\alpha\beta} + q$, we have

$$iN \sum_{\alpha\beta} \hat{q}_{\alpha\beta} q_{\alpha\beta} = \frac{nN}{2} [\Delta\hat{q} + \hat{q}_1(1-q)] + \mathcal{O}(n^2). \tag{19}$$

and

$$-i \sum_{\alpha\beta} \hat{q}_{\alpha\beta} \sum_i w_i^\alpha w_i^\beta = -\frac{1}{2}(\Delta\hat{q} + \hat{q}_1) \sum_\alpha \sum_i (w_i^\alpha)^2 - \frac{1}{2}\hat{q}_1 \sum_{(\alpha\beta)} \sum_i w_i^\alpha w_i^\beta$$

$$= -\frac{1}{2}\Delta\hat{q} \sum_\alpha \sum_i (w_i^\alpha)^2 - \frac{1}{2}\hat{q}_1 \sum_i \left( \sum_\alpha w_i^\alpha \right)^2 \tag{20}$$

$$\overset{\text{HST}}{=} -\frac{1}{2}\Delta\hat{q} \sum_\alpha \sum_i (w_i^\alpha)^2 + \sqrt{-\hat{q}_1} \sum_i t_i \left( \sum_\alpha w_i^\alpha \right),$$

where in the last step HST denotes Hubbard-Stratonovich transformation $\int \frac{dt}{\sqrt{2\pi}} e^{-t^2/2} e^{bt} = e^{b^2/2}$ that we use to linearize the quadratic term at the cost of introducing an auxiliary Gaussian variable $t$ to be averaged over later.

Recall that $\hat{q}(k) = \int e^{ikw}\hat{p}(w) = \frac{1}{N} \sum_i^N e^{ikw_i^\alpha}$, the distribution constraint becomes

$$\delta\left( \int dk\, (\hat{q}(k) - q(k)) \right) = \delta\left( \int dk \left( \frac{1}{N} \sum_i^N e^{ikw_i^\alpha} - q(k) \right) \right)$$

$$= \int \frac{d\hat{\lambda}_\alpha(k)}{2\pi} \exp\left\{ \int dk\, i\hat{\lambda}_\alpha(k) \left( \sum_i e^{ikw_i^\alpha} - Nq(k) \right) \right\}. \tag{21}$$

Note that $\sum_i \int dk\, i\hat{\lambda}_\alpha(k) e^{ikw_i^\alpha} = 2\pi i \sum_i \lambda_\alpha(-w_i^\alpha)$ by inverse Fourier transform. Next,

$$-iN \int dk\, \hat{\lambda}_\alpha(k) q(k) = -iN \int dk \left( \int dw\, e^{ikw} \lambda_\alpha(w) \right) \left( \int dw'\, e^{ikw'} q(w') \right)$$

$$= -2\pi iN \int dw\, dw'\, \lambda_\alpha(w) q(w') \delta(w + w') \tag{22}$$

$$= -2\pi iN \int dw\, q(w) \lambda_\alpha(-w).$$

Now we can write down the full free energy. We ignore overall constant coefficients such as $2\pi$'s and $i$'s in the integration measure, which become irrelevant upon taking the saddle-point approximation. We also leave out the denominator of $V$, as it does not depend on data and is an overall constant. Note that under the replica-symmetric ansatz the replica index $\alpha$ gives $n$ identical copies of the same integral and thus the replica index $\alpha$ can be suppressed (same for synaptic index $i$):

$$\langle V^n \rangle = \int dq d\hat{\lambda}(k) d\Delta\hat{q} d\hat{q}_1 e^{nN(G_0+G_1)}, \tag{23}$$

where (please note that $q$ is replica overlap, and $q(w)$ is the imposed target distribution)

$$G_0 = \frac{1}{2}\Delta\hat{q} + \frac{1}{2}\hat{q}_1(1-q) - 2\pi i \int dw q(w)\lambda(-w) + \langle \log Z(t) \rangle_t,$$
$$Z(t) = \int dw \exp\left\{ 2\pi i \lambda(-w) - \frac{1}{2}\Delta\hat{q} w^2 + \sqrt{-\hat{q}_1} tw \right\}. \tag{24}$$

Note that integrals in Eqn.23 can be evaluated using saddle-point approximation in the thermodynamic limit $N \to \infty$.

Redefining $2\pi i \lambda(-w) - \frac{1}{2}\Delta\hat{q} w^2 \to -\lambda(w)$ and $-\hat{q}_1 \to \hat{q}_1$, we have

$$G_0 = \frac{1}{2}\Delta\hat{q} - \frac{1}{2}\hat{q}_1(1-q) + \int dw q(w)\lambda(w) - \frac{1}{2}\Delta\hat{q} \int dw q(w)w^2 + \langle \log Z(t) \rangle_t,$$
$$Z(t) = \int dw \exp\left\{ -\lambda(w) + \sqrt{\hat{q}_1} tw \right\}. \tag{25}$$

We seek the saddle-point solution for $G_0$ with respect to the order parameters $\Delta\hat{q}$, $\lambda(w)$, and $\hat{q}_1$:

$$0 = \frac{\partial G_0}{\partial \Delta\hat{q}} \Rightarrow 1 = \int dw q(w)w^2 = \langle w^2 \rangle_{q(w)}, \tag{26}$$

$$0 = \frac{\partial G_0}{\partial \lambda(w)} \Rightarrow q(w) = \left\langle \frac{1}{Z(t)} \exp\left\{ -\lambda(w) + \sqrt{\hat{q}_1} tw \right\} \right\rangle. \tag{27}$$

We observe that the saddle-point equation Eqn.26 fixes the second moment of the imposed distribution $q(w)$ to 1 and therefore can be thought of as a second moment constraint. $G_0$ now simplifies to

$$G_0 = -\frac{1}{2}\hat{q}_1(1-q) + \int dw q(w)\lambda(w) + \langle \log Z(t) \rangle_t. \tag{28}$$

The remaining $\hat{q}_1$ saddle-point equation is a bit more complicated,

$$0 = \frac{\partial G_0}{\partial \hat{q}_1} = -\frac{1}{2}(1-q) + \frac{t}{2\sqrt{\hat{q}_1}} \left\langle \frac{1}{Z(t)} \int dw w \exp\left\{ -\lambda(w) + \sqrt{\hat{q}_1} tw \right\} \right\rangle_t \tag{29}$$

Integration by parts for the second term in rhs:

$$1 - q = \frac{1}{\sqrt{\hat{q}_1}} \int Dt \frac{1}{Z} \sqrt{\hat{q}_1} \int dw w^2 \exp\left\{ -\lambda(w) + \sqrt{\hat{q}_1} tw \right\}$$
$$- \frac{1}{\sqrt{\hat{q}_1}} \int Dt \frac{1}{Z^2} \sqrt{\hat{q}_1} \left( \int dw w \exp\left\{ -\lambda(w) + \sqrt{\hat{q}_1} tw \right\} \right)^2 \tag{30}$$
$$= \left\langle \langle w^2 \rangle_{f(w)} \right\rangle_t - \left\langle \langle w \rangle_{f(w)}^2 \right\rangle_t,$$

where in the last step we have defined an induced distribution $f(w) = Z(t)^{-1} \exp\left\{-\lambda(w) + \sqrt{\hat{q}_1}tw\right\}$. Since the second moments are fixed to 1, we have

$$q = \left\langle \langle w \rangle_{f(w)}^2 \right\rangle_t, \tag{31}$$

which gives a nice interpretation of $q$ in terms of the average overlap of $w$ in the induced distribution $f(w)$.

**Limit $q \to 1$**

We are interested in the critical load $\alpha_c$ where the version space (space of viable weights) shrinks to a single point, i.e., there exists only one viable solution. Since $q$ measures the typical overlap between weight vectors in the version space, the uniqueness of the solution implies $q \to 1$ at $\alpha_c$. In this limit, the order parameters $\{\hat{q}_1, \lambda(w)\}$ diverges and we need to re-derive the saddle point equations Eqn.27 and Eqn.31 in terms of the undiverged order parameters $\{u, r(w)\}$:

$$\hat{q}_1 = \frac{u^2}{(1-q)^2}; \qquad \lambda(w) = \frac{r(w)}{1-q}. \tag{32}$$

Now $G_0$ becomes

$$G_0 = \frac{1}{1-q}\left\{-\frac{1}{2}u^2 + \int dw q(w)r(w) + (1-q)\left\langle \log Z(t) \right\rangle_t\right\}, \tag{33}$$

and

$$Z(t) = \int dw \exp \frac{1}{1-q}\left\{-r(w) + utw\right\}. \tag{34}$$

We can perform a saddle-point approximation for the $w$ integral in $Z(t)$ at the saddle value $w$ such that $r'(w) = ut$:

$$Z(t) = \exp\left\{\frac{-r(w) + utw}{1-q}\right\}. \tag{35}$$

Then

$$G_0 = \frac{1}{1-q}\left\{-\frac{1}{2}u^2 + \int dw q(w)r(w) - \left\langle r(w) \right\rangle_t + u \left\langle tw \right\rangle\right\}. \tag{36}$$

Let's use integration by parts to rewrite

$$\int dw q(w)r(w) = -\int Q(w)r'(w)dw$$
$$\left\langle r(w) \right\rangle_t = \int \frac{dt}{\sqrt{2\pi}}e^{-t^2/2}r(w) = -\int P(t)r'(w)dw, \tag{37}$$

where $Q(w)$ is the CDF of the imposed distribution $q(w)$ and $P(t) = \frac{1}{2}\left[1 + \text{Erf}(\frac{t}{\sqrt{2}})\right]$ is the normal CDF.

Now the saddle-point equation

$$0 = \frac{\partial G_0}{\partial r'(w)} \Rightarrow Q(w) = P(t) \tag{38}$$

determines $w(t)$ implicitly. The $u$ equation gives

$$0 = \frac{\partial G_0}{\partial u} \Rightarrow u = \langle tw \rangle_t = \left\langle \frac{dw}{dt} \right\rangle_t \tag{39}$$

where in the last equality we have used integration by parts. Using Eqn.38-39 $G_0$ is simplified to

$$G_0 = \frac{1}{2(1-q)} \left\langle \frac{dw}{dt} \right\rangle_t^2. \tag{40}$$

**The energetic part**

We would like to perform a similar procedure as shown above, to Eqn.17 using the replica-symmetric ansatz. We observe that the effect of the distribution constraint is entirely captured in $G_0$ and therefore $G_1$ is unchanged compared with the standard Gardner calculation of perceptron capacity. We reproduce the calculation here for completeness.

Under the replica-symmetric ansatz $q_{\alpha\beta} = (1-q)\delta_{\alpha\beta} + q$, Eqn.17 becomes

$$\begin{aligned}
K &= i \sum_\alpha x_\alpha \rho_\alpha - \frac{1-q}{2} \sum_\alpha x_\alpha^2 - \frac{q}{2} \left( \sum_\alpha x_\alpha \right)^2 \\
&\stackrel{\text{HST}}{=} i \sum_\alpha x_\alpha \rho_\alpha - \frac{1-q}{2} \sum_\alpha x_\alpha^2 - it\sqrt{q} \sum_\alpha x_\alpha.
\end{aligned} \tag{41}$$

where we have again used the Hubbard-Stratonovich transformation to linearize the quadratic piece. Performing the Gaussian integrals in $x_\alpha$ (define $\alpha = \frac{P}{N}$),

$$nG_1 = \alpha \log \left[ \left\langle \int_\kappa^\infty \frac{d\rho}{\sqrt{2\pi(1-q)}} \exp\left\{ -\frac{(\rho + t\sqrt{q})^2}{2(1-q)} \right\} \right\rangle_t^n \right]. \tag{42}$$

At the limit $n \to 0$,

$$nG_1 = \alpha n \left\langle \log \left[ \int_\kappa^\infty \frac{d\rho}{\sqrt{2\pi(1-q)}} \exp\left\{ -\frac{(\rho + t\sqrt{q})^2}{2(1-q)} \right\} \right] \right\rangle_t. \tag{43}$$

Perform the Gaussian integral in $\rho$ and define $\tilde{\kappa} = \frac{\kappa + t\sqrt{q}}{\sqrt{1-q}}$, we have

$$G_1 = \alpha \int Dt \log H(\tilde{\kappa}). \tag{44}$$

At the limit $q \to 1, \alpha \to \alpha_c, \int_{-\infty}^\infty Dt$ is dominated by $\int_{-\kappa}^\infty Dt$, and $H(\tilde{\kappa}) \to \frac{1}{\sqrt{2\pi}\tilde{\kappa}} e^{-\tilde{\kappa}^2/2}$. The $\mathcal{O}\left(\frac{1}{1-q}\right)$ (leading order) contribution gives

$$G_1 = -\frac{1}{2(1-q)} \alpha_c \int_{-\kappa}^\infty Dt(\kappa + t)^2. \tag{45}$$

Let $G = G_0 + G_1$. As $n \to 0$, $\langle V^n \rangle = e^{n(NG)} \to 1 + n(NG)$, and $\langle \log V \rangle = \lim_{n \to 0} \frac{\langle V^n \rangle - 1}{n} = NG$.

Combining with Eqn.40 (relabel $t \leftrightarrow x$ to distinguish between the two auxiliary Gaussian variables), we have

$$\langle \log V \rangle = \frac{N}{2(1-q)} \left[ \left\langle \frac{dw}{dx} \right\rangle_x^2 - \alpha_c \int_{-\kappa}^{\infty} Dt(\kappa+t)^2 \right] \tag{46}$$

Capacity $\alpha_c$ is reached when Eqn.13 goes to zero. We arrive at the distribution-constrained capacity

$$\alpha_c(\kappa) = \alpha_0(\kappa) \left\langle \frac{dw}{dx} \right\rangle_x^2, \tag{47}$$

where $\alpha_0(\kappa) = \left[ \int_{-\kappa}^{\infty} Dt(\kappa+t)^2 \right]^{-1}$ is the unconstrained capacity.

### Instructive Examples

(1) Standard normal distribution $w \sim \mathcal{N}(0,1)$.

In this case $w = x$ and $\alpha_c(\kappa) = \alpha_0(\kappa)$.

(2) Normal distribution with nonzero mean $w \sim \mathcal{N}(\mu, \sigma^2)$. This is the example discussed in the main text Fig.1.

In this case $w = \mu + \sigma x$ and $\mu^2 + \sigma^2 = 1$ due to the second moment constraint Eqn.26. Then $\alpha_c(\kappa) = \sigma^2 \alpha_0(\kappa)$.

(3) Lognormal distribution $w \sim \frac{1}{\sqrt{2\pi}w} \exp\left\{ -\frac{(\ln w - \mu)^2}{2\sigma^2} \right\}$.

In this case $w = e^{\mu + \sigma x}$ where $\mu = -\sigma^2$. $\alpha_c(\kappa) = \sigma^2 e^{-\sigma^2} \alpha_0(\kappa)$.

### Geometrical interpretation

Note that although the Jacobian factor $\left\langle \frac{dw}{dx} \right\rangle_x$ takes a simple form, in practice sometimes it might not be the most convenient form to use. Integrating by parts ($p(x) = \mathcal{N}(0,1)$),

$$\left\langle \frac{dw}{dx} \right\rangle_x = \int dx p(x) wx \tag{48}$$

Now define $u = P(x)$ so that $du = p(x)dx$ and $w = Q^{-1}(P(x)) = Q^{-1}(u)$, we can express the Jacobian in terms of the CDFs

$$\left\langle \frac{dw}{dx} \right\rangle_x = \int_0^1 du \left( Q^{-1}(u) P^{-1}(u) \right) \tag{49}$$

Furthermore,

$$\left\langle \frac{dw}{dx} \right\rangle_x = \frac{1}{2} \left[ \int_0^1 du \left( Q^{-1}(u) \right)^2 + \int_0^1 du \left( P^{-1}(u) \right)^2 - \int_0^1 du \left( Q^{-1}(u) - P^{-1}(u) \right)^2 \right] \tag{50}$$
$$= \frac{1}{2} \left[ 2 - W_2(P,Q)^2 \right],$$

where we have used second moments equal to $1$ and the definition of the Wasserstein-$k$ distance in the second equality. Therefore, we have arrived at the geometric interpretation of the Jacobian term

$$\left\langle \frac{dw}{dx} \right\rangle_x = 1 - \frac{1}{2} W_2(P,Q)^2. \tag{51}$$

### A.1.2 Theory for an arbitrary number of synaptic subpopulations

In this section, we generalize our theory in the above section to the set up of a perceptron with $M$ synaptic populations indexed by $m$, $\boldsymbol{w}^m$, such that each $w_i^m$ satisfies its own distributions constraints $w_i^m \sim q_m(w^m)$. We denote the overall weight vector as $\boldsymbol{w} \equiv \{\boldsymbol{w}^m\}_{m=1}^M \in \mathbb{R}^{N \times 1}$, where the total number of weights is $N = \sum_{m=1}^M N_m$. The replica overlap now becomes $q_{\alpha\beta} = \frac{1}{N} \sum_m^M \sum_i^{N_m} w_i^{m\alpha} w_i^{m\beta}$. The distribution constraint becomes (see Eqn.21 for the case of $M = 1$)

$$\prod_m \delta \left( \int dk^m \left( \frac{1}{N_m} \sum_i^{N_m} e^{ik^m w_i^{m\alpha}} - q_m(k^m) \right) \right). \tag{52}$$

We introduce $\hat{q}_{\alpha\beta}, \lambda_m(k)$ to write the $\delta$-functions into Fourier representations, and use replica-symmetric ansatz $\hat{q}_{\alpha\beta} = -\frac{i}{2}(\Delta\hat{q}\delta_{\alpha\beta} + \hat{q}_1)$, and $q_{\alpha\beta} = (1-q)\delta_{\alpha\beta} + q$ as before. After similar manipulations that lead to Eqn.25, the entropic part of the free energy becomes ($g_m = N_m/N$ is the fraction of weights in $m$-th population)

$$G_0 = \frac{1}{2}\Delta\hat{q} - \frac{1}{2}\hat{q}_1(1-q) + \sum_m g_m \int dw^m q_m(w^m)\lambda_m(w^m)$$
$$- \frac{1}{2}\Delta\hat{q} \sum_m g_m \int dw^m q_m(w^m)(w^m)^2 + \sum_m g_m \langle \log Z_m(t) \rangle_t, \tag{53}$$
$$Z_m(t) = \int dw^m \exp\left\{ -\lambda_m(w^m) + \sqrt{\hat{q}_1} t w^m \right\}.$$

Now the second moment constraint $0 = \partial G_0/\partial\Delta\hat{q}$ (Eqn.26) becomes the weighted sum of second moments from each population:

$$1 = \sum_m g_m \int dw^m q_m(w^m)(w^m)^2 = \sum_m g_m \left\langle (w^m)^2 \right\rangle_{q_m}. \tag{54}$$

We take the $q \to 1$ limit as before:

$$\hat{q}_1 = \frac{u^2}{(1-q)^2}; \qquad \lambda_m(w^m) = \frac{r_m(w^m)}{1-q}. \tag{55}$$

Use saddle-point approximation for $Z_m(t)$ and integrate by parts as in Eqn.35-37, the entropic part becomes

$$G_0 = \frac{1}{1-q}\left\{ -\frac{1}{2}u^2 + \sum_m g_m r_m'(w^m)\left[ P(x) - Q_m(w^m) \right] + u \sum_m g_m \langle t w^m \rangle_t \right\}. \tag{56}$$

Now the saddle-point equation for order parameters $r_m'(w^m)$ and $u$ gives

$$P(x) = Q_m(w^m)$$
$$u = \sum_m g_m \langle t w^m \rangle_t = \sum_m g_m \left\langle \frac{dw^m}{dt} \right\rangle_t. \tag{57}$$

Therefore,

$$G_0 = \frac{1}{2(1-q)}\left[ \sum_m g_m \left\langle \frac{dw^m}{dt} \right\rangle_t \right]^2. \tag{58}$$

The energetic part (Eqn.35) remains unchanged and thus (relabel $t \leftrightarrow x$)

$$\alpha_c(\kappa) = \alpha_0(\kappa) \left[ \sum_m g_m \left\langle \frac{dw^m}{dx} \right\rangle_x \right]^2.$$ (59)

**E/I balanced lognormals**

Now we specialize to the biologically realistic E/I balanced lognormal distributions described in the main text. We are interested the case with two synaptic populations $m = E, I$ that models the excitatory/inhibitory synpatic weights of a biological neuron. $w_i^E \sim \frac{1}{\sqrt{2\pi}\sigma_E w^E} \exp \left\{ -\frac{(\ln w^E - \mu_E)^2}{2\sigma_E^2} \right\}$ and $w_i^I \sim \frac{1}{\sqrt{2\pi}\sigma_I w^I} \exp \left\{ -\frac{(\ln w^I - \mu_I)^2}{2\sigma_I^2} \right\}$. Let's denote the E/I fractions as $g_E = r$ and $g_I = 1 - r$. The CDF of the lognormals are given by

$$Q_m(w^m) = H \left[ \frac{1}{\sigma_m} (\mu_m - \ln w^m) \right].$$ (60)

The corresponding inverse CDF is

$$Q_m^{-1}(u) = \exp \left\{ \mu_m - \sigma_m H^{-1}(u) \right\}.$$ (61)

The capacity is therefore

$$\alpha_c = \alpha_0 \left[ \sum_m g_m \int_0^1 du Q_m^{-1}(u) P^{-1}(u) \right]^2$$
$$= \alpha_0 \left[ r \int_0^1 du H^{-1}(u) \exp \left\{ \mu_E - \sigma_E H^{-1}(u) \right\} + (1-r) \int du H^{-1}(u) \exp \left\{ \mu_I - \sigma_I H^{-1}(u) \right\} \right]^2.$$ (62)

This model has five parameters $\{r, \sigma_E, \sigma_I, \mu_E, \mu_I\}$. We use values of $r$ reported in experiments (the ratio between of E. connections found and I. connections found).

We also have two constraints. The E/I balanced constraint $g_E \left\langle w^E \right\rangle_{q_E} = g_I \left\langle w^I \right\rangle_{q_I}$:

$$re^{\mu_E + \frac{1}{2}\sigma_E^2} = (1-r)e^{\mu_I + \frac{1}{2}\sigma_I^2},$$ (63)

and the second moment constraint $1 = \sum_m g_m \left\langle (w^m)^2 \right\rangle_{q_m}$:

$$1 = re^{2(\mu_E + \sigma_E^2)} + (1-r)e^{2(\mu_I + \sigma_I^2)}.$$ (64)

Therefore there are two free parameters left and we choose to express $\mu_E$ and $\mu_I$ in terms of the rest:

$$\mu_I = -\frac{1}{2}\sigma_I^2 - \ln(1-r) - \frac{1}{2} \ln \left[ \frac{e^{\sigma_I^2}}{1-r} + \frac{e^{\sigma_E^2}}{r} \right]$$
$$\mu_E = -\frac{1}{2}\sigma_E^2 - \ln r - \frac{1}{2} \ln \left[ \frac{e^{\sigma_I^2}}{1-r} + \frac{e^{\sigma_E^2}}{r} \right].$$ (65)

The parameter landscape is plotted against the two free parameters $\sigma_E$ and $\sigma_I$. Here we report comparisons across different experiments [38, 9, 32, 47, 65, 74] similar to main text Fig.4 (Fig.4 (a) is included here for reference). We estimate $\sigma_E$ and $\sigma_I$ using the experimentally reported coefficient of variation $(CV)$ as it is a dimensionless quantity suitable for comparison across different experiments.

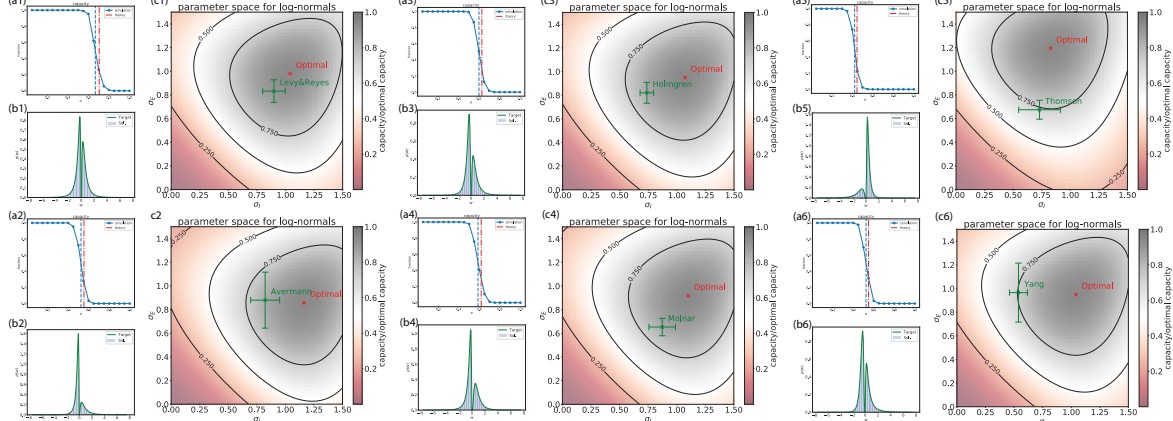

Figure 6: Additional parameter landscape for the biologically-realistic distribution. (a)-(b) (theory from main text Eqn.10 and simulations from DisCo-SGD): (a) Determination of capacity; (b) Example of weight distribution obtained in simulation. (c) Capacity (normalized by the optimal value in the landscape) as a function of the lognormal parameters $\sigma_E$ and $\sigma_I$. Experimental value is shown in green with error bars, and optimal capacity is shown in red.

We then estimate $\sigma$ assuming the underlying experimental E. or I. distributions are lognormal, such that $CV = \sqrt{e^{\sigma^2} - 1}$. See e.g. Table S2 in the supplementary materials of [18] for a complied list of $CV$s for different experiments.

Note that despite the apparently different shape of distributions, all the experimentally measured parameter values are within the first quantile of the optimal values predicted by our theory.

### A.1.3 Capacity for biased inputs and sparse label

In this section, we generalized our theory in Section A.1.1 to the set up of nonzero-mean input patterns $\boldsymbol{\xi}^\mu$ and sparse labels $\zeta^\mu$:

$$
\begin{aligned}
p(\xi_i^\mu) &= \mathcal{N}(m, 1 - m^2) \\
p(\zeta^\mu) &= f\delta(\zeta^\mu - 1) + (1 - f)\delta(\zeta^\mu + 1).
\end{aligned}
\tag{66}
$$

In this case, we need to include a bias in the perceptron $\hat{\zeta}^\mu = \text{sgn}(\frac{\boldsymbol{w}\cdot\boldsymbol{\xi}^\mu}{\|\boldsymbol{w}\|} - b)$ to be able to correctly classify patterns in general.

Note that $m = 0$ and $f = 1/2$ reduces to the case in Section A.1.1. We observe due to the multiplicative relation between the Jacobian term and the original Gardner capacity in Eqn.47, entropic effects (such as distribution constraints and sign-constraints) factors with the energetic effects (such as the nonzero mean inputs and sparse labels), and they don't interfere with each other. Therefore, the calculations for nonzero mean inputs and sparse labels are identical with the original Gardner case. Here we only reproduce the calculation for completeness. Readers already familiar with this calculation should skip this part.

The analog of Eqn.15 reads (define the local fields as $h_i^\mu = \sum_\alpha x_\alpha^\mu w_i^\alpha$)

$$\prod_{\mu\alpha}\left\langle e^{-\frac{i}{\sqrt{N}}x_\alpha^\mu\zeta^\mu\boldsymbol{\xi}^\mu\cdot\boldsymbol{w}^\alpha}\right\rangle_{\xi\zeta}=\prod_{\mu i}\left\langle\exp\left\{-\frac{i}{\sqrt{N}}\zeta^\mu\xi_i^\mu h_i^\mu\right\}\right\rangle_{\xi\zeta}$$

$$=\prod_{\mu i}\left\langle\exp\left\{-\frac{im}{\sqrt{N}}\zeta^\mu h_i^\mu-\frac{1}{2N}(1-m^2)(h_i^\mu)^2\right\}\right\rangle_\zeta \qquad (67)$$

$$=\prod_\mu\left\langle\exp\left\{-im\zeta^\mu\sum_\alpha x_\alpha^\mu M_\alpha-\frac{1-m^2}{2}\sum_{\alpha\beta}x_\alpha^\mu x_\beta^\mu q_{\alpha\beta}\right\}\right\rangle_\zeta,$$

where in the second equality we have carried out the Gaussian integral in $\boldsymbol{\xi}^\mu$ and in the third equality we introduced the order parameters

$$q_{\alpha\beta}=\frac{1}{N}\sum_i w_i^\alpha w_i^\beta,\qquad M_\alpha=\frac{1}{\sqrt{N}}\sum_i w_i^\alpha. \qquad (68)$$

Now the full energetic term becomes

$$\left\langle\Theta\left(\frac{1}{\sqrt{N}}\zeta^\mu\boldsymbol{\xi}^\mu\cdot\boldsymbol{w}^\alpha-b\zeta^\mu-\kappa\right)\right\rangle_{\xi\zeta}$$

$$=\prod_\mu\left\langle\int_{\kappa+b\zeta^\mu}^\infty\frac{d\lambda_\alpha^\mu}{2\pi}\int dx_\alpha^\mu\exp\left\{-im\zeta^\mu\sum_\alpha x_\alpha^\mu M_\alpha-\frac{1-m^2}{2}\sum_{\alpha\beta}x_\alpha^\mu x_\beta^\mu q_{\alpha\beta}\right\}\right\rangle_\zeta$$

$$=f\prod_\mu\int_{\kappa+b}^\infty\frac{d\lambda_\alpha^\mu}{2\pi}\int dx_\alpha^\mu\exp\left\{i\sum_\alpha x_\alpha^\mu(\lambda_\alpha^\mu-mM_\alpha)-\frac{1-m^2}{2}\sum_{\alpha\beta}x_\alpha^\mu x_\beta^\mu q_{\alpha\beta}\right\}$$

$$+(1-f)\prod_\mu\int_{\kappa-b}^\infty\frac{d\lambda_\alpha^\mu}{2\pi}\int dx_\alpha^\mu\exp\left\{i\sum_\alpha x_\alpha^\mu(\lambda_\alpha^\mu+mM_\alpha)-\frac{1-m^2}{2}\sum_{\alpha\beta}x_\alpha^\mu x_\beta^\mu q_{\alpha\beta}\right\}$$

$$=f\prod_\mu\int_{\frac{\kappa+b-mM_\alpha}{\sqrt{1-m^2}}}^\infty\frac{d\lambda_\alpha^\mu}{2\pi}\int dx_\alpha^\mu\exp\left\{i\sum_\alpha x_\alpha^\mu\lambda_\alpha^\mu-\frac{1}{2}\sum_{\alpha\beta}x_\alpha^\mu x_\beta^\mu q_{\alpha\beta}\right\}$$

$$+(1-f)\prod_\mu\int_{\frac{\kappa-b+mM_\alpha}{\sqrt{1-m^2}}}^\infty\frac{d\lambda_\alpha^\mu}{2\pi}\int dx_\alpha^\mu\exp\left\{i\sum_\alpha x_\alpha^\mu\lambda_\alpha^\mu-\frac{1}{2}\sum_{\alpha\beta}x_\alpha^\mu x_\beta^\mu q_{\alpha\beta}\right\}.$$

Now $G_1$ becomes

$$G_1=\frac{1}{1-q}\left\{f\int_{\frac{\kappa-b+mM}{\sqrt{1-m^2}}}^\infty Dt\left(t+\frac{\kappa+b-mM}{\sqrt{1-m^2}}\right)^2+(1-f)\int_{\frac{-\kappa-b-mM}{\sqrt{1-m^2}}}^\infty Dt\left(t+\frac{\kappa-b+mM}{\sqrt{1-m^2}}\right)^2\right\}. \qquad (69)$$

Note that the hat-variables $\hat{M}_\alpha$ conjugated with $M_\alpha$ are in subleading order to $\hat{q}_{\alpha\beta}$ in the thermodynamic limit, and therefore $G_0$ is unchanged. Let $v=M-b/m$, we have now the capacity

$$\alpha_c(\kappa)=\left\langle\frac{dw}{dx}\right\rangle_x^2\left[f\int_{\frac{-\kappa+mv}{\sqrt{1-m^2}}}^\infty Dt\left(t+\frac{\kappa-mv}{\sqrt{1-m^2}}\right)^2+(1-f)\int_{\frac{-\kappa-mv}{\sqrt{1-m^2}}}^\infty Dt\left(t+\frac{\kappa+mv}{\sqrt{1-m^2}}\right)^2\right]^{-1}, \qquad (70)$$

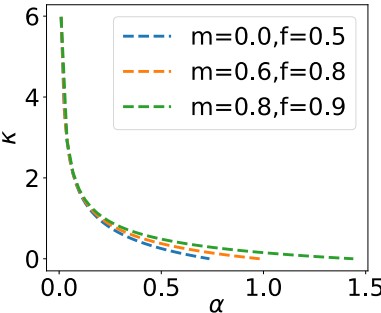

Figure 7: $\alpha_c(\kappa)$ for different values of input mean $m$ and label sparsity $f$. Note that the blue curve corresponds to the vanilla case shown in main text Fig.4(c).

where the order parameter $v$ needs to be determined from the saddle-point equation

$$f \int_{\frac{-\kappa+mv}{\sqrt{1-m^2}}}^{\infty} Dt \left(t + \frac{\kappa - mv}{\sqrt{1-m^2}}\right) = (1-f) \int_{\frac{-\kappa-mv}{\sqrt{1-m^2}}}^{\infty} Dt \left(t + \frac{\kappa + mv}{\sqrt{1-m^2}}\right). \tag{71}$$

In Fig.7 we numerically solve $\alpha_c(\kappa)$ for different values of $m$ and $f$.

## A.2 Optimal transport theory

In recent years, Wasserstein distances has found diverse applications in fields ranging from machine learning [7, 26, 49] to geophysics [23, 24, 19, 45, 46]. In optimal transport theory, the Wasserstein-$k$ distance arise as the minimal cost one needs to pay in transporting one probability distribution to another, when the moving cost between probability masses are measured by the $L_k$ norm [72]. When one equips the probability density manifold with the Wasserstein-2 distance as metric, it becomes the Wasserstein space, a Riemannian manifold of real-valued distributions with a constant nonnegative sectional curvature [41, 25, 20]. Note that in our statistical mechanical theory main text Eqn.3-5, the Wasserstein-2 distance naturally arises in the mean-field limit without assuming any a priori transportation cost.

Here we briefly review the theory of optimal transport. Intuitively, optimal transport concerns the problem of finding the shortest path of morphing one distribution into another. In the following, we will use the *Monge* formulation [66, 3].

Given probability distributions $P$ and $Q$ with supports $X$ and $Y$, we say that $T : X \to Y$ is a transport map from $P$ to $Q$ if the *push-forward* of $P$ through $T$, $T_\# P$, equals $Q$:

$$Q = T_\# P \equiv P(T^{-1}(Y)). \tag{72}$$

Eqn.72 can be understood as moving probability masses $x \in X$ from distribution $P$ to $y \in Y$ according to transportation map $T$, such that upon completion the distribution over $Y$ becomes $Q$.

We are interested in finding a transportation plan that minimizes the transportation cost as measured by some distance function $d : X \times Y \to \mathbb{R}$:

$$C(T; d) = \int_X d(T(x), x) p(x) dx \qquad \text{s.t. } T_\# P = Q. \tag{73}$$

The transportation plan that minimizes Eqn.73 is called the optimal transport plan $T^* = \text{argmin}_T C(T; d)$. When the distance function $d$ is chosen to be the $L_k$ norm, the minimal cost becomes the Wasserstein-$k$ distance:

$$W_k(P, Q) = \inf_T C(T; L_k)|_{T_\# P = Q}. \tag{74}$$

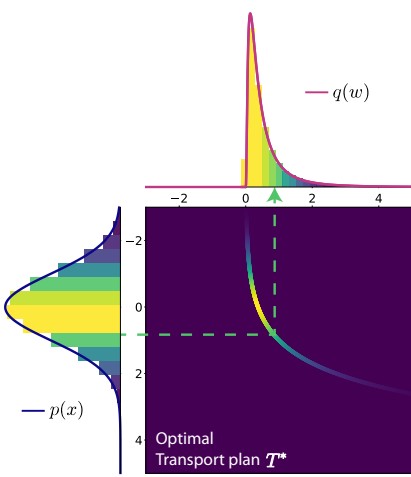

Figure 8: An example optimal transport plan from standard normal, $p(x)$, to a lognormal distribution $q(w)$. The optimal transport plan $T^*$ is plotted in between the distributions. $T^*$ moves $p(x)$ units of probability mass $x$ to location $w$, as indicated by the dashed line, and the colors are chosen to reflect the amount of probability mass to be transported.

In $1$-dimension, the Wasserstein-$k$ distance has a closed form given by main text Eqn.6, and the optimal transport map has an explicit formula in terms of the CDFs: $T^* = Q^{-1} \circ P$. An example of the optimal transport map and how it moves probability masses between distributions is given in Fig.8 for transport between $p(w) = \mathcal{N}(0, 1)$ and $q(w) = \frac{1}{\sqrt{2\pi}\sigma w} \exp\left\{\frac{(\ln w - \mu)^2}{2\sigma^2}\right\}$. Note that in this case, the optimal transport plan is simply $T^*(x) = e^{\mu + \sigma x}$.

Now consider the manifold $\mathcal{M}$ of real-valued probability distributions, where points on this manifold are probability measures that admits a probability density function. When endowed with the $W_k$ metric, $(\mathcal{M}, W_k)$ becomes a metric space and is in particular a geodesic space [66, 3]. We can explicitly construct the geodesics connecting points on $\mathcal{M}$. We parameterize the geodesic by the *geodesic time* $\tau \in [0, 1]$. Then given $T^*$ an optimal transport plan, the intermediate probability distributions along the geodesic take the following form [66]:

$$P_\tau = ((1 - \tau)\text{Id} + \tau T^*)_{\#} P \tag{75}$$

where Id is the identity map and $P_\tau$ is a constant speed geodesic connecting $P_{\tau=0} = P$ and $P_{\tau=1} = Q$.

For the discrete case, we can describe the sample $\{w_i^\tau\}$ from $P_\tau$ in a simple manner in terms of the samples $\{w_i\}$ drawn from $P$ and $\{\hat{w}_i\}$ drawn from $Q$. We can arrange the samples in the ascending order, or equivalently, forming their order statistics $\{x_{(i)} : x_{(1)} \le ... \le x_{(N)}\}$, which can be thought of as atoms in a discrete measure. Then in terms of the order statistics,

$$w_{(i)}^\tau = (1 - \tau)w_{(i)} + \tau\hat{w}_{(i)} \tag{76}$$

Upon infinitetesimal change in the geodesic time, $\tau \to \tau + \delta\tau$, the geodesic flow becomes

$$w_{(i)}^{\tau+\delta\tau} = w_{(i)}^\tau + \delta\tau \left(\hat{w}_{(i)} - w_{(i)}\right) \tag{77}$$

Specializing to the case discussed in main text Section 3, $w_{(i)} = w_{(i)}^{\tau=0}$ is the initialization for the perceptron weight and therefore just a constant, we can promoted it $w_{(i)} \to w_{(i)}^\tau$ to fix the overall scale in the perceptron weight, then we arrive at main text Eqn.9.

## A.3 Generalization supplemental materials

### A.3.1 Replica calculation of generalization with sign-constraint

In this section, we calculate the sign-constraint teacher-student setup. To ease notation, let's denote the teacher perceptron $\boldsymbol{w_t} \equiv \boldsymbol{w}^0$ and the (replicated) student perceptron $\boldsymbol{w_s^a} \equiv \boldsymbol{w^a}$. Given random inputs $\boldsymbol{\xi}^\mu$ with $p(\xi_i^\mu) = \mathcal{N}(0,1)$, we generate labels by $\zeta^\mu = \text{sgn}(\boldsymbol{w}^0 \cdot \boldsymbol{\xi}^\mu / ||\boldsymbol{w}^0|| + \eta^\mu)$, where $\eta^\mu$ is input noise and $\eta^\mu \sim \mathcal{N}(0, \sigma^2)$. Let's denote the signs of the teacher perceptron as $s_i = \text{sgn}(w_i^0)$. The student perceptron's weights are constrained to have the same sign as that of the teacher's, so we insert $\Theta(s_i w_i^a)$ in the Gardner volume to enforce this constraint (we leave out the denominator part of $V$ as it does not depend on data and is an overall constant):

$$\langle V^n \rangle_{\xi\eta w^0} = \prod_{\alpha=1}^{n} \left\langle \int_{-\infty}^{\infty} \frac{d\boldsymbol{w}^a}{\sqrt{2\pi}} \prod_{\mu=1}^{p} \Theta\left(\text{sgn}\left(\frac{\boldsymbol{w}^0 \cdot \boldsymbol{\xi}^\mu}{||\boldsymbol{w}^0||} + \eta^\mu\right) \frac{\boldsymbol{w^a} \cdot \boldsymbol{\xi}^\mu}{||\boldsymbol{w^a}||} - \kappa\right) \prod_{i}^{N} \Theta(s_i w_i^a) \right\rangle_{\xi\eta w^0}.$$

$$(78)$$

We observe that upon redefining $s_i w_i^a \rightarrow w_i^a, s_i \xi_i^\mu \rightarrow \xi_i^\mu$, we can absorb the sign-constraints into the integration range of $w$ from $[-\infty, +\infty]$ to $[0, \infty]$:

$$\langle V^n \rangle_{\xi\eta w^0} = \prod_{\alpha=1}^{n} \left\langle \int_{0}^{\infty} \frac{d\boldsymbol{w}^a}{\sqrt{2\pi}} \prod_{\mu=1}^{p} \Theta\left(\text{sgn}\left(\frac{\boldsymbol{w}^0 \cdot \boldsymbol{\xi}^\mu}{||\boldsymbol{w}^0||} + \eta^\mu\right) \frac{\boldsymbol{w^a} \cdot \boldsymbol{\xi}^\mu}{||\boldsymbol{w^a}||} - \kappa\right) \right\rangle_{\xi\eta w^0}. \qquad (79)$$

Therefore, sign constraint amounts to restricting all the weights to be positive. In the following, we denote $\int_0^\infty$ as $\int_c$.

Let's define the local fields as

$$h_\mu^a = \frac{\boldsymbol{w^a} \cdot \boldsymbol{\xi}^\mu}{\sqrt{N}}; \qquad h_\mu^0 = \frac{\boldsymbol{w^0} \cdot \boldsymbol{\xi}^\mu}{\sqrt{N}} + \eta^\mu \qquad (80)$$

We leave the average over teacher $w^0$ to the very end.

$$\langle V^n \rangle_{\xi\eta} = \prod_{\mu a} \int_c \frac{d\boldsymbol{w}^a}{\sqrt{2\pi}} \int dh_\mu^a \Theta\left(\text{sgn}(h_\mu^0) h_\mu^a - \kappa\right) \left\langle \delta\left(h_\mu^a - \frac{\boldsymbol{w^a} \cdot \boldsymbol{\xi}^\mu}{\sqrt{N}}\right) \right\rangle_{\xi\eta}$$

$$= \int_c \left(\prod_{a=1}^{n} \frac{d\boldsymbol{w}^a}{\sqrt{2\pi}}\right) \int \prod_{\mu a} \frac{dh_\mu^a d\hat{h}_\mu^a}{2\pi} \int \prod_{\mu} \frac{dh_\mu^0 d\hat{h}_\mu^0}{2\pi} \prod_{\mu a} \Theta\left(\text{sgn}(h_\mu^0) h_\mu^a - \kappa\right)$$

$$\times \left\langle \exp\left\{ \sum_{\mu a}\left(i\hat{h}_\mu^a h_\mu^a - i\hat{h}_\mu^a \frac{\boldsymbol{w^a} \cdot \boldsymbol{\xi}^\mu}{\sqrt{N}}\right) + \sum_{\mu}\left(i\hat{h}_\mu^0 h_\mu^0 - i\hat{h}_\mu^0 \frac{\boldsymbol{w^0} \cdot \boldsymbol{\xi}^\mu}{\sqrt{N}} - i\hat{h}_\mu^0 \eta^\mu\right)\right\}\right\rangle_{\xi\eta}$$

$$= \int_c \left(\prod_{a=1}^{n} \frac{d\boldsymbol{w}^a}{\sqrt{2\pi}}\right) \int \prod_{\mu a} \frac{dh_\mu^a d\hat{h}_\mu^a}{2\pi} \int \prod_{\mu} \frac{dh_\mu^0 d\hat{h}_\mu^0}{2\pi} \prod_{\mu a} \Theta\left(\text{sgn}(h_\mu^0) h_\mu^a - \kappa\right)$$

$$\times \exp\left\{\sum_{\mu a} i\hat{h}_\mu^a h_\mu^a + \sum_{\mu} i\hat{h}_\mu^0 h_\mu^0\right\}$$

$$\times \prod_{\mu} \exp\left\{ -\frac{1}{2N}\left[\sum_{a,b} \hat{h}_\mu^a \hat{h}_\mu^b \sum_{i} w_i^a w_i^b + N\left(\hat{h}_\mu^0\right)^2 + 2\sum_{a} \hat{h}_\mu^a \hat{h}_\mu^0 \sum_{i} w_i^a w_i^0\right]\right\},$$

$$(81)$$

where in the last step we perform the average over noise $\eta^\mu \sim \mathcal{N}(0, \sigma^2)$ and patterns $p(\xi_i^\mu) = \mathcal{N}(0,1)$, and make use of the normalization conditions $\sum_i (w_i^0)^2 = N$ and $\sum_i (w_i^a)^2 = N$.

Now let's define order parameters

$$q_{ab} = \frac{1}{N}\sum_i w_i^a w_i^b, \qquad R_a = \frac{1}{N}\sum_i w_i^a w_i^0. \tag{82}$$

We introduce conjugate variables $\hat{q}_{ab}$ and $\hat{R}_a$ to write the $\delta$-functions into its Fourier representations, and after some algebraic manipulations we can bring the Gardner volume into the following form ($\alpha \equiv p/N$):

$$\langle\langle V^n\rangle\rangle_{\xi,z} = \int(\prod_a d\hat{q}_1^a)(\prod_{ab} dq^{ab}d\hat{q}^{ab})(\prod_a dR^a d\hat{R}^a)e^{nNG}, \tag{83}$$

where ($\bar{h}_\mu^0 = \gamma h_\mu^0; \quad \gamma = 1/\sqrt{1+\sigma^2}$)

$$nG = nG_0 + \alpha nG_E$$
$$nG_0 = -\frac{1}{2}\sum_{ab}\hat{q}^{ab}q^{ab} - \sum_a \hat{R}^a R^a + n\langle \ln Z\rangle_{w^0},$$
$$Z = \int_c \left(\prod_a \frac{dw_i^a}{\sqrt{2\pi}}\right)\exp\left\{\frac{1}{2}\sum_a \hat{q}_1^a(w_i^a)^2 + \frac{1}{2}\sum_{a\neq b}\hat{q}^{ab}w_i^a w_i^b + \sum_a \hat{R}^a w_i^a w_i^0\right\},$$
$$nG_1 = \ln\int\prod_a \frac{d\hat{h}^a dh^a}{2\pi}\int D\bar{h}^0 \prod_a \Theta\left(\mathrm{sgn}(\frac{\bar{h}^0}{\gamma})h^a - \kappa\right)$$
$$\times \exp\left\{i\sum_a \hat{h}^a h^a - i\gamma\bar{h}^0\sum_a h^a R^a - \frac{1}{2}\sum_a (\hat{h}^a)^2[1-(\gamma R^a)^2] - \frac{1}{2}\sum_{a\neq b}\hat{h}^a\hat{h}^b(q^{ab}-\gamma^2 R^a R^b)\right\}. \tag{84}$$

The energetic part $G_1$ is the same as the unconstrained case in [61, 22]. After standard manipulations, we have

$$G_1 = 2\int DtH\left(-\frac{\gamma Rt}{\sqrt{q-\gamma^2 R^2}}\right)\ln H\left(\frac{\kappa-\sqrt{q}t}{\sqrt{1-q}}\right). \tag{85}$$

**Entropic part**

In this subsection, we perform the integrals in the entropic part, and we will see novel terms coming from the constraint on the student's integration range.

We start by assuming a replica-symmetric solution for the auxiliary variables introduced in the Fourier decomposition of the $\delta$-functions,

$$\hat{R}^a = \hat{R}; \qquad \hat{q}^{ab} = \hat{q} + (\hat{q}_1 - \hat{q})\delta_{ab}; \qquad \hat{q}_1^a = \hat{q}_1; \qquad m_i^a = m_i; \qquad \hat{m}_i^a = \hat{m}_i, \tag{86}$$

and $q_{ab} = (1-q)\delta_{ab} + q$.

Then the entropic part,

$$Z = \int\left(\prod_a \frac{dw_i^a}{\sqrt{2\pi}}\right)\exp\left\{\frac{1}{2}(\hat{q}_1-\hat{q})\sum_a (w_i^a)^2 + \hat{R}w_i^0\sum_a w_i^a + \frac{1}{2}\hat{q}(\sum_a w_i^a)^2\right\}$$
$$\stackrel{\mathrm{HST}}{=}\int Dt\int_c(\prod_a \frac{dw_i^a}{\sqrt{2\pi}})\exp\left\{\frac{1}{2}(\hat{q}_1-\hat{q})\sum_a (w_i^a)^2 + (\hat{R}w_i^0 + t\sqrt{\hat{q}})\sum_a w_i^a\right\}, \tag{87}$$

where we have introduced Gaussian variable $t$ to linearize quadratic term as usual. Now the integral becomes $n$ identical copies and we can drop the replica index $a$,

$$G_0 = -\frac{1}{2}\hat{q}_1 + \frac{1}{2}\hat{q}q - \hat{R}R + \langle \ln Z \rangle_{t,w^0}. \tag{88}$$

We can bring the log term into the form of an induced distribution $f(w)$,

$$Z = \int_0^\infty \frac{dw}{\sqrt{2\pi}} \exp\left[-f(w)\right]$$
$$f(w) = \frac{1}{2}(\hat{q} - \hat{q}_1)w^2 - (\hat{R}w^0 + t\sqrt{\hat{q}})w \tag{89}$$

Under saddle-point approximation, we obtain a set of mean field self-consistency equations for the order parameters:

$$0 = \frac{\partial G_0}{\partial \hat{q}_1} \Rightarrow 1 = \left\langle \langle w^2 \rangle_f \right\rangle_{t,w^0}$$
$$0 = \frac{\partial G_0}{\partial \hat{R}} \Rightarrow R = \left\langle w^0 \langle w \rangle_f \right\rangle_{t,w^0}, \tag{90}$$
$$0 = \frac{\partial G_0}{\partial \hat{q}} \Rightarrow q = \left\langle \langle w \rangle_f^2 \right\rangle_{t,w^0}$$

$$0 = \frac{\partial G_1}{\partial q} \Rightarrow \hat{q} = -2\alpha\partial_q G_1$$
$$0 = \frac{\partial G_1}{\partial R} \Rightarrow \hat{R} = \alpha\partial_R G_1 \tag{91}$$

$q \to 1$ **limit**

In this limit the order parameter diverges, and we define the new set of undiverged order parameters as

$$\hat{R} = \frac{\tilde{R}}{1-q}; \qquad \hat{q} = \frac{\tilde{q}^2}{(1-q)^2}; \qquad \hat{q} - \hat{q}_1 = \frac{\Delta}{1-q}. \tag{92}$$

Then

$$f(w) = \frac{1}{1-q}\left[\frac{1}{2}\Delta w^2 - (\tilde{R}w^0 + t\tilde{q})w\right]$$
$$= \frac{1}{1-q}\left[\frac{1}{2}\Delta\left(w - \frac{1}{\Delta}(\tilde{R}w^0 + t\tilde{q})\right)^2 - \frac{1}{2\Delta}(\tilde{R}w^0 + t\tilde{q})^2\right]. \tag{93}$$

Then $\langle w \rangle_f = \frac{1}{\Delta}\left(\tilde{R}w^0 + t\tilde{q}\right)$, and the integral over the auxiliary variable is dominated by values of $t$ such that $\tilde{R}w^0 + t\tilde{q} > 0$. In the following, we denote $\left\langle [g(t)]_+ \right\rangle_t$ as integrating over range of $t$ such that $g(t) > 0$. Then the self-consistency equations Eqn.90 take a compact form (after rescaling order parameters $\tilde{R} \to \tilde{R}\Delta$, $\tilde{q} \to \tilde{q}\Delta$)

$$1 = \frac{1}{\Delta} \left\langle \Theta(\tilde{R}w^0 + t\tilde{q}) \right\rangle_{t,w^0}$$

$$1 = \left\langle \left[\tilde{R}w^0 + t\tilde{q}\right]_+^2 \right\rangle_{t,w^0} \quad , \tag{94}$$

$$R = \left\langle w^0 \left[\tilde{R}w^0 + t\tilde{q}\right]_+ \right\rangle_{t,w^0}$$

Eqn.90 becomes ($\tilde{\kappa} = \kappa/\sqrt{1 - \gamma^2 R^2}$)

$$\tilde{R}\Delta = \frac{\alpha\gamma}{\sqrt{2\pi}} \sqrt{1 - \gamma^2 R^2} \int_{-\tilde{\kappa}}^{\infty} Dt \left(\tilde{\kappa} + t\right)$$

$$\frac{\Delta}{2} \left(2 - \tilde{q}^2\Delta - 2\tilde{R}R\right) = \alpha \int_{-\infty}^{\kappa} DtH \left(-\frac{\gamma Rt}{\sqrt{1 - \gamma^2 R^2}}\right) (\kappa - t)^2 \tag{95}$$

The free energy is (recall that $\gamma = 1/\sqrt{1 + \sigma^2}$)

$$G = \frac{1}{2(1-q)} \left(\Delta - \tilde{q}^2 - 2\tilde{R}R + \frac{1}{\Delta} \left\langle \left[\tilde{R}w^0 + t\tilde{q}\right]_+^2 \right\rangle_{t,w^0}\right) - \alpha \int_{-\infty}^{\kappa} DtH \left(-\frac{\gamma Rt}{\sqrt{1 - \gamma^2 R^2}}\right) (\kappa - t)^2. \tag{96}$$

### A.3.2 Replica calculation of generalization with distribution-constraint

In this subsection, we will consider the case where student weights are constrained to some *prior* distribution $q_s(w_s)$, while the teacher obeys a distribution $p_t(w_t)$, for an arbitrary pair $q_s, p_t$. We can write down the Gardner volume $V_g$ for generalization as in the capacity case (main text Eqn.2):

$$V_g = \frac{\int d\boldsymbol{w}_s \left[\prod_{\mu=1}^P \Theta \left(\text{sgn}\left(\frac{\boldsymbol{w}_t \cdot \boldsymbol{\xi}^\mu}{||\boldsymbol{w}_t||} + \eta^\mu\right) \frac{\boldsymbol{w}_s \cdot \boldsymbol{\xi}^\mu}{||\boldsymbol{w}_s||} - \kappa\right)\right] \delta(||\boldsymbol{w}_s||^2 - N)\delta\left(\int dk \left(\hat{q}(k) - q(k)\right)\right)}{\int d\boldsymbol{w}_s \delta(||\boldsymbol{w}_s||^2 - N)}. \tag{97}$$

We treat the distribution constraint $q_s(w)$ similar to Section A.1.1. The entropic part of the free energy becomes

$$G_0 = -\frac{1}{2}\hat{q}_1 + \frac{1}{2}\hat{q}q - \hat{R}R + \int_{-\infty}^{\infty} dw q_s(w)\lambda(w) + \langle \ln Z \rangle_{t,w_t}$$

$$Z = \int \frac{dw}{\sqrt{2\pi}} \exp\left[-f(w)\right] \tag{98}$$

$$f(w) = \frac{1}{2}(\hat{q} - \hat{q}_1)w^2 - (\hat{R}w_t + t\sqrt{\hat{q}})w + \lambda(w)$$

At the limit $q \to 1$, we make the following ansatz

$$\hat{R} = \frac{\tilde{R}}{1-q}; \quad \hat{q} = \frac{u^2}{(1-q)^2}; \quad \hat{q} - \hat{q}_1 = \frac{\Delta}{1-q}; \quad \lambda(w) = \frac{r(w)}{1-q}. \tag{99}$$

Then

$$G_0 = \frac{1}{(1-q)} \left( -\frac{1}{2}u^2 + \frac{1}{2}\Delta - \tilde{R}R + \int dw q_s(w) r(w) \right) + \langle \ln Z \rangle_{t,w_t}$$

$$f(w) = \frac{1}{1-q} \left( \frac{1}{2}\Delta w^2 - (\tilde{R}w_t + ut)w + r(w) \right) \tag{100}$$

We can absorb $\frac{1}{2}\Delta w^2$ into the definition of $r(w)$, $\frac{1}{2}\Delta w^2 + r(w) \to r(w)$, and $0 = \partial G_0/\partial \Delta$ gives the second moment constraint, $1 = \int dw q_s(w) w^2$.

Then,

$$G_0 = \frac{1}{(1-q)} \left( -\frac{1}{2}u^2 - \tilde{R}R + \int dw q(w) r(w) \right) + \langle \ln Z \rangle_{t,w_t}$$

$$f(w) = \frac{1}{1-q} \left( r(w) - (\tilde{R}w_t + ut)w \right) \tag{101}$$

Next, we perform a saddle-point approximation on the log-term in $G_0$,

$$Z = \int \frac{dw}{\sqrt{2\pi}} \exp\left[ -f(w) \right] \approx \exp\left[ -f(w_s) \right], \tag{102}$$

where $w_s$ is the saddle-point value for the weight, and is determined implicitly by

$$r'(w_s) = \tilde{R}w_t + ut. \tag{103}$$

Note that $r'(w_s)$ is now an induced random variable from random variables $w_t$ and $t$. For later convenience, we rescale $r'(w_s)$ to define a new random variable $z$,

$$z \equiv u^{-1} r'(w_s) = t + u^{-1}\tilde{R}w_t \equiv t + \varepsilon w_t, \tag{104}$$

where we have also defined

$$\varepsilon \equiv u^{-1}\tilde{R}. \tag{105}$$

The induced distribution on $z$ is then

$$\tilde{p}(z) = \int Dt \int dw_t p(w_t) \delta(z - t - \varepsilon w_t). \tag{106}$$

Now the entropic part becomes

$$G_0 = \frac{1}{(1-q)} \left( -\frac{1}{2}u^2 - \tilde{R}R + \int dw q_s(w) r(w) + \langle (\tilde{R}w_t + ut)w_s \rangle_{t,w_t} - \langle r(w_s) \rangle_{t,w_t} \right). \tag{107}$$

Integrate by parts,

$$\int dw q(w) r(w) = - \int dw Q(w) r'(w), \tag{108}$$

$$\begin{aligned}
\langle r(w_s) \rangle_{t,w_t} &= \int Dt dw_t p_t(w_t) r(w_s) \\
&= \int dz \delta(z - t - \varepsilon w_t) \int Dt dw_t p_t(w_t) r(w_s) \\
&= \int dz \tilde{p}(z) r(w_s) \\
&= - \int dz \tilde{P}(z) r'(w_s)
\end{aligned} \tag{109}$$

Now $0 = \partial G/\partial r'(w_s)$ gives

$$Q(w_s) = \tilde{P}(z). \tag{110}$$

which implicitly determines $w_s(z)$.

Next,

$$0 = \frac{\partial G}{\partial u} \Rightarrow u = \langle w_s(z)t \rangle_{t,w_t}, \tag{111}$$

$$0 = \frac{\partial G}{\partial \tilde{R}} \Rightarrow R = \langle w_s(z)w_t \rangle_{t,w_t}. \tag{112}$$

The free energy then simplifies to

$$G = \frac{u^2}{2(1-q)} + \alpha G_1. \tag{113}$$

The energetic part as $q \to 1$ becomes (same as the unconstrained and sign-constrained case)

$$G_1 = -\frac{1}{1-q} \int_{-\infty}^{\kappa} DtH\left(-\frac{\gamma Rt}{\sqrt{1-\gamma^2 R^2}}\right)(\kappa - t)^2. \tag{114}$$

The remaining two saddle point equations are (1) the vanishing log-Gardner volume and (2) $0 = \partial G/\partial R$:

$$\frac{1}{2}u^2 = \alpha \int_{-\infty}^{\kappa} DtH\left(-\frac{\gamma Rt}{\sqrt{1-\gamma^2 R^2}}\right)(\kappa - t)^2, \tag{115}$$

$$\varepsilon u = \alpha\gamma\sqrt{\frac{2}{\pi}}\sqrt{1-\gamma^2 R^2} \int_{-\tilde{\kappa}}^{\infty} Dt\left(\tilde{\kappa} + t\right). \tag{116}$$

In summary, the order parameters $\{R, \kappa, u, \varepsilon\}$ can be determined from a set of self-consistency equations:

$$
\begin{aligned}
u &= \langle w_s(z)t \rangle_{t,w_t} \\
R &= \langle w_s(z)w_t \rangle_{t,w_t} \\
\frac{1}{2}u^2 &= \alpha \int_{-\infty}^{\kappa} DtH\left(-\frac{\gamma Rt}{\sqrt{1-\gamma^2 R^2}}\right)(\kappa - t)^2, \\
\varepsilon u &= \frac{2\alpha\gamma}{\sqrt{2\pi}}\sqrt{1-\gamma^2 R^2} \int_{-\tilde{\kappa}}^{\infty} Dt\left(\tilde{\kappa} + t\right)
\end{aligned}
\tag{117}
$$

where we have introduced $\tilde{\kappa} = \kappa/\sqrt{1-\gamma^2 R^2}$, an auxiliary normal variable $t \sim \mathcal{N}(0,1)$, and an induced random variable $z \equiv t + \varepsilon w_t$ with induced distribution

$$\tilde{p}(z) = \int Dt \int dw_t p_t(w_t)\delta(z - t - \varepsilon w_t). \tag{118}$$

Note that $w_s(z)$ can be determined implicitly by equating the CDF of the induced variable $z$ and the distribution that the student is constrained to:

$$Q(w_s) = \tilde{P}(z). \tag{119}$$

**Examples**

(1) Lognormal distribution

In the following, we solve $w_s(z)$ explicitly from the CDF equation $Q(w_s) = \tilde{P}(z)$. For a lognormal teacher,

$$p_t(w_t) = \frac{1}{w_t}\frac{1}{\sqrt{2\pi}\sigma}\exp\left\{-\frac{(\ln w_t - \mu)^2}{2\sigma^2}\right\}. \tag{120}$$

The second moment constraint implies $\mu = -\sigma^2$.

The induced CDF of $z$ is

$$\tilde{P}(z) = \int_{-\infty}^{z} dz' \int_{-\infty}^{\infty} Dt \int_{0}^{\infty} dw_t p_t(w_t) \delta(z' - t - \varepsilon w_t). \tag{121}$$

Let $x = (\ln w - \mu)/\sigma$,

$$\tilde{P}(z) = \int_{-\infty}^{z} dz' \int_{-\infty}^{\infty} Dt \int_{-\infty}^{\infty} Dx \delta(z' - t - \varepsilon e^{\mu+\sigma x})$$
$$= \int_{-\infty}^{\infty} Dx H(\varepsilon e^{\mu+\sigma x} - z) \tag{122}$$

Now the CDF of $w_s$ is

$$Q_s(w_s) = \int_{-\infty}^{w_s} q_s(w) dw = H\left(-\frac{\ln w_s - \mu}{\sigma}\right). \tag{123}$$

Therefore, equating $\tilde{P}(z)$ and $Q_s(w_s)$:

$$\int_{-\infty}^{\infty} Dx H(\varepsilon e^{\mu+\sigma x} - z) = H\left(-\frac{\ln w_s - \mu}{\sigma}\right), \tag{124}$$

We can solve for $w_s(z)$ by (recall $z \equiv t + \varepsilon w_t$)

$$w_s(z) = \exp\left\{\mu + \sigma H^{-1}\left(\int Dx H(z - \varepsilon e^{\mu+\sigma x})\right)\right\}. \tag{125}$$

Or in terms of error functions

$$w_s(z) = \exp\left\{\mu + \sqrt{2}\sigma \mathrm{erf}^{-1}\left(\int Dx \mathrm{erf}\left(\frac{\varepsilon e^{\mu+\sigma x} - z}{\sqrt{2}}\right)\right)\right\}. \tag{126}$$

We can also calculate the initial overlap (before any learning):

$$R_0 = \langle \boldsymbol{w}_t \cdot \boldsymbol{w}_s \rangle_{p_t q_s} = e^{2\mu+\sigma^2} = e^{-\sigma^2}. \tag{127}$$

(2) Uniform distribution

Assuming that both the teacher and the student have a uniform distribution in range $[0, \sigma]$.

The second moment constraint fixes $\sigma = \sqrt{3}$.

We can solve (as in the lognormal example above),

$$w_s(z) = \frac{1}{\varepsilon} \int_{-\infty}^{z} dz' \left(H(z' - \varepsilon\sigma) - H(z')\right). \tag{128}$$

(3) Half-normal distribution

Assuming that both the teacher and the student has a half-normal distribution $\frac{2}{\sqrt{2\pi}\sigma} \exp\left\{-\frac{w^2}{2\sigma^2}\right\}$.

The second moment constraint fixes $\sigma = 1$, and

$$w_s(z) = \sigma H^{-1}\left\{\frac{1}{2} - \int_{-\infty}^{\frac{z}{\sqrt{1+\sigma^2\varepsilon^2}}} Dt H(-\sigma\varepsilon t)\right\}. \tag{129}$$

**Arbitrary number of synaptic subpopulations**

Just like in the case of Section A.1.2, we can generalize our theory above to incorporate distribution constraints with an arbitrary number of synaptic subpopulations. Let's consider a student perceptron with $M$ synaptic populations indexed by $m$, $\boldsymbol{w}^m$, such that each $w_i^m$ satisfies its own distributions constraints $w_i^m \sim Q_m(w^m)$. We denote the overall weight vector as $\boldsymbol{w} \equiv \{\boldsymbol{w}^m\}_{m=1}^M \in \mathbb{R}^{N \times 1}$. The total number of weights is $N = \sum_{m=1}^M N_m$, and we denote the fractions as $g_m = N_m/N$. Since the derivation is similar to that of Section A.1.2 and Section A.3.2, we will only present the results here.

As before, the order parameters $\{R, \kappa, u, \varepsilon\}$ can be determined from a set of self-consistency equations:

$$
\begin{aligned}
u &= \sum_m g_m \langle w^m(z)t \rangle_{t,w_t} \\
R &= \sum_m g_m \langle w^m(z)w_t \rangle_{t,w_t} \\
\frac{1}{2}u^2 &= \alpha \int_{-\infty}^{\kappa} DtH\left(-\frac{\gamma R t}{\sqrt{1-\gamma^2 R^2}}\right)(\kappa - t)^2 \\
\varepsilon u &= \frac{2\alpha\gamma}{\sqrt{2\pi}}\sqrt{1-\gamma^2 R^2}\int_{-\tilde{\kappa}}^{\infty} Dt\left(\tilde{\kappa}+t\right)
\end{aligned}
\tag{130}
$$

where $\tilde{\kappa} = \kappa/\sqrt{1-\gamma^2 R^2}$, $t \sim \mathcal{N}(0,1)$. and an induced random variable $z \equiv t + \varepsilon w_t$ with induced distribution the same as Eqn.118.

Note that every $w^m(z)$ can be determined by equating the CDF of the induced variable $z$ and the $m$-th distribution that $w^m(z)$ is constrained to:

$$
Q_m(w^m) = \tilde{P}(z).
\tag{131}
$$

### A.3.3 Sparsification of weights in sign-constraint learning

For unconstrained weights, max-margin solutions are considered beneficial for generalization particularly for small size training sets. As a first step toward biological plausibility, one can try to constraint the sign of individual weights during learning (e.g., excitatory or inhibitory). In the generalization error setup, we can impose a constraint that the teacher and student have the same set of weight signs. Surprisingly, we find both analytically and numerically that if the teacher weights are not too sparse, the max-margin solution generalizes poorly: after a single step of learning (with random input vectors), the overlap, $R$, drops substantially from its initial value $R_0$ (by a factor of $\sqrt{2}$ for a half-Gaussian teacher, see the blue curves in Fig.9(a).

We can verify this by calculating $R_0$ in two different ways. As an example, in the following we consider the case where both the teacher and student have half-normal distributions.

(1) By definition, the overlap is $R = \frac{\boldsymbol{w}_s \cdot \boldsymbol{w}_t}{\|\boldsymbol{w}_s\|\|\boldsymbol{w}_t\|}$. Since $\boldsymbol{w}_s$ and $\boldsymbol{w}_t$ are uncorrelated before learning ($\alpha = 0$), the initial overlap is then $R_0 = \frac{\langle w_s \rangle \langle w_t \rangle}{\|\boldsymbol{w}_s\|\|\boldsymbol{w}_t\|} = \frac{2}{\pi}$;

(2) Take the $\alpha \to 0$ limit in Eqn.90 and Eqn.91 and calculate $R_{0+} = \lim_{\alpha \to 0+} R(\alpha) = \frac{\sqrt{2}}{\pi}$.

Therefore, in this example $R_{0+} = R_0/\sqrt{2}$.

The source of the problem is that due to the sign constraint, max-margin training with few examples yields a significant mismatch between the student and teacher weight distributions. After only a few steps of learning, half of the student's weights are set to zero, and the student's distribution, $p(w_s) = \frac{1}{2}\delta(0) + \frac{1}{\sqrt{2\pi}}\exp\{-\frac{w_s^2}{4}\}$, deviates significantly from the teacher's half-normal distribution (Fig.9(b)).

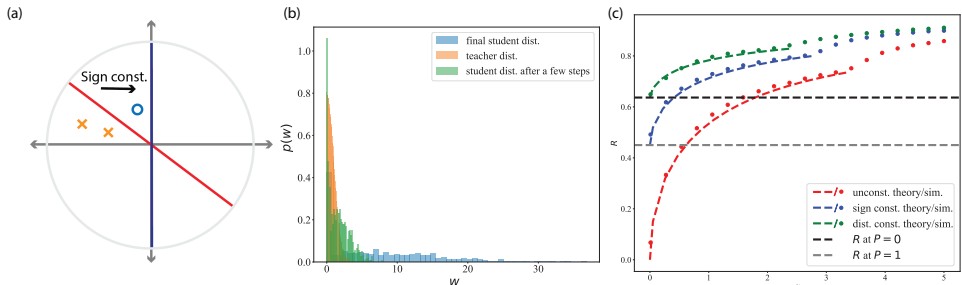

Figure 9: Sparsification of weights in sign-constraint learning. (a) An illustration of weight sparsification. In this schematic, the perceptron lives on this 1-dimensional circle and $N = 2$. Red line denotes the hyperplane orthogonal to the perceptron weight before sign-constraint, crosses and circles indicate examples in different classes. Sign-constraint pushes the weights to the first quadrant, which zeros half of the weights on average. Blue line indicates the hyperplane obtained after the sign-constraint. (b) Sparsification of weights due to max-margin training. After only a few iterations, nearly half of the student weights are set to zero, and the distribution deviates significantly from the teacher's distribution. (c) Teacher-student overlap as a function of load $\alpha$ for different learning paradigms. Dashed lines are from theory, and dots are from simulation. Note the horizontal dashed lines show the initial drop in overlap from zero example and to just a single example. In this case teacher has nonzero noise, $\gamma = 0.85$.

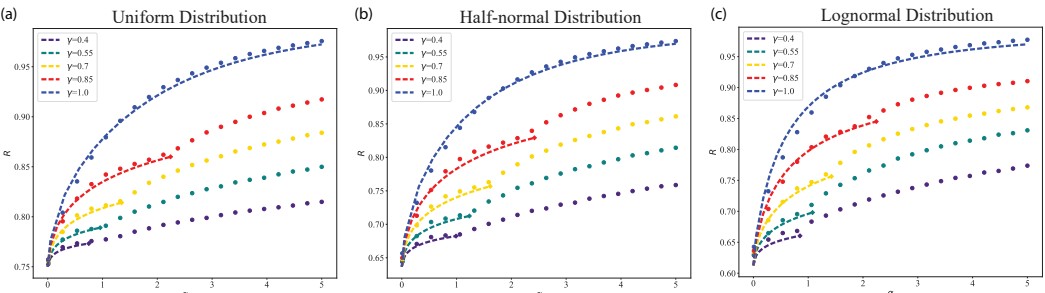

Figure 10: Generalization (measured by overlap) performance for different distributions and different noise levels in fixed prior learning. From left to right: uniform, half-normal, and lognormal distribution. In all cases the student is constrained to have the same distribution as that of the teacher's. Dashed lines are from theory and dots are from DisCo-SGD simulation.

### A.3.4 Noisy teacher

We generate examples $\{\boldsymbol{\xi}^\mu, \zeta^\mu\}_{\mu=1}^P$ from a teacher perceptron, $\boldsymbol{w}_t \in \mathbb{R}^N$: $\zeta^\mu = \text{sgn}(\boldsymbol{w}_t \cdot \boldsymbol{\xi}^\mu / ||\boldsymbol{w}_t|| + \eta^\mu)$, where $\eta^\mu$ is input noise and $\eta^\mu \sim \mathcal{N}(0, \sigma^2)$. In this subsection we present additional numerical results for the case when $\sigma \neq 0$. As in previous sections, we define the noise level parameter $\gamma = 1/\sqrt{1 + \sigma^2}$.

Our theory's prediction is confirmed by numerical simulation for a wide range of teacher noise level $\gamma$ and teacher weight distributions $P_t(w_t)$. We find that distribution-constrained learning performs consistently better all the way up to capacity (capacity in this framework is due to teacher noise). For illustration, in Fig.10 we show theory and simulation for fixed prior learning of three different teacher distributions: uniform, half-normal, and lognormal.

### A.4 DisCo-SGD simulations

**Avoid vanishing gradients**

Note that we often observe a vanishing gradient in DisCo-SGD when we choose a constant learning rate $\eta_1$, and in such cases the algorithm tends to find poor margin $\kappa$ which deviates from the max-

margin value predicted from the theory. We find that scaling $\eta_1$ with the standard deviation of the gradient solves this problem:

$$\eta_1 = \eta_1^0/\text{std}\left(\sum_\mu \xi_i^\mu(\hat{\zeta}^\mu - \zeta^\mu)\right), \tag{132}$$

where the standard deviation is computed across the synaptic index $i$ and $\eta_1^0$ is a constant.

**Mini-batches**

For the capacity simulations, we always use full-batch in the SGD update, so it is in fact simply gradient descent. However, in the case of generalization, we find that training with mini-batches improves the generalization performance, since it acts as an source of stochasticity during training. In main text Fig.5 we use mini-batch size $B = 0.8P$ (80% of examples are used for each SGD update).

When we vary teacher's noise level, we find that scaling $B$ with $\gamma$ improves the quality of the solutions, as measured by the generalization performance (or equivalently, the teacher-student overlap). Generally, the more noisy the teacher is, the smaller the mini-batches should be. This is because smaller mini-batch size corresponds to higher stochasticity, which helps overcoming higher teacher noise.

**Parameters**

All the capacity simulations are performed with the following parameters $N = 1000, \eta_1^0 = 0.01, \eta_2 = 0.6, t_{max} = 10000$, where $t_{max}$ is the maximum number of iterations of the DisCo-SGD algorithm.

All results are averaged over 300 realizations.

In main text Fig.4, the experimental [38] parameters are $g_E = 45.8\%, \sigma_E = 0.833, \sigma_I = 0.899$.

In main text Fig.5(a): We show the teacher-student overlap as a function of $\alpha$. Dots are simulations performed with series of student distribution from $\sigma_s = 0.1$ to $\sigma_s = 1.4$, where the teacher distribution sits in the middle of this range, $\sigma_t = 0.7$. Each such simulation is performed with fixed $\sigma_s$ and varying load $\alpha \in [0.05, 2.5]$. In main text Fig.5(b): we show the empirical weight distributions found by unconstrained perceptron learning for $\alpha \in [0.05, 10]$. In main text Fig.5(c) we show optimal student distribution for $\alpha \in [0.05, 2.5]$. Note that optimal prior learning approaches the teacher distribution much faster than unconstrained learning.

All the generalization DisCo-SGD simulations are performed with the same parameter as in the capacity DisCo-SGD simulations, but with two additional parameter: teacher's noise level $\gamma$ and SGD mini-batch size $B$.

For the simulations in Fig.10 we use

$\gamma = 0.4, B = 0.2P; \gamma = 0.55, B = 0.4P; \gamma = 0.7, B = 0.6P; \gamma = 0.85, B = 0.8P; \gamma = 1.0, B = P$ (noiseless case).

## A.5 Replica symmetry breaking

### A.5.1 Bimodal distributions

In deriving the capacity formula, we have assumed replica-symmetry (RS). It is well-known that replica-symmetry breaking occurs in the Ising perceptron [52, 13], so it is natural to ask to what extent our theory holds when approaching the Ising limit. Let's consider a bimodal distribution with a mixture of two normal distributions with non-zero mean centered around zero,

$$p(w) = \frac{1}{2}\mathcal{N}(-\mu, \sigma) + \frac{1}{2}\mathcal{N}(\mu, \sigma)$$

The second moment constraint requires $\mu^2 + \sigma^2 = 1$.

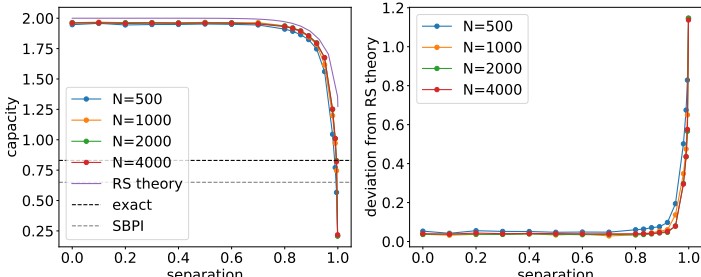

Figure 11: Left: Capacity as a function of separation for different size perceptrons. Dots are from DisCo-SGD simulations and the 'RS theory' line is from our theory. Exact values for Ising perceptron and state-of-the-art numerical values are included as well. Right: Deviation from the RS theory as a function of separation. This is the same as subtracting the simulation values from the theoretical predictions in the left figure.

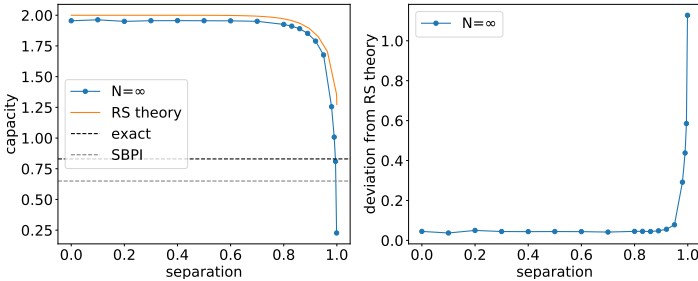

Figure 12: Finite size effects. Left/Right: we extrapolate simulation values in Fig.11 Left/Right to infinite $N$.

We can gradually decrease the Gaussian width $\sigma$, or equivalently $\mu = \sqrt{1 - \sigma^2}$ (which we call 'separation' in the following) and compare the capacity theoretically predicted by the RS theory and numerically found by the DisCo-SGD algorithm.

In Fig.11 we can see that the simulation agrees well with the RS theory until one gets very close to the Ising limit ($\mu = 1$). To understand finite size effects, we extrapolate to the infinite size limit ($N \to \infty$) in Fig.12, and found that the deviation from RS theory has a sharp transition near $\mu = 1$, marking the breakdown of the RS theory.

**Ising perceptron**

It is also interesting to compare our distribution-constrained RS theory to the unconstrained RS theory. In this Ising limit,

$$q(w) = \frac{1}{2}\delta(w - 1) + \frac{1}{2}\delta(w + 1), \tag{133}$$

and CDF

$$Q(w) = \frac{1}{2}\Theta(w - 1) + \frac{1}{2}\Theta(w + 1). \tag{134}$$

Equating $Q(w)$ with the normal CDF $P(x)$ and solve for $w(x)$, we find $w(x) = \text{sgn}(x)$. Then $dw/dx = 2\delta(x)$ and $\left\langle \frac{dw}{dx} \right\rangle_x = \frac{2}{\sqrt{2\pi}}$. Therefore,

$$\lim_{Ising} \alpha_c(\kappa = 0) = \frac{4}{\pi}, \tag{135}$$

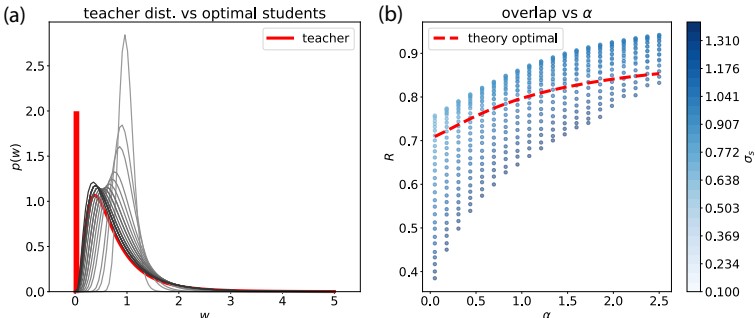

Figure 13: Optimal student prior distribution as a function of $\alpha$. (a) Gray curves correspond to a series of optimal student distributions as a function of $\alpha$, with the darker color representing larger $\alpha$. Red is teacher distribution. (b) Overlap as a function of $\alpha$ for different student priors. Red dashed line is the optimal overlap calculated from our replica-symmetric theory. Dots are from DisCo-SGD simulations. For the same $\alpha$, different color dots represent different overlaps obtained from simulations with different $\sigma_s$.

which is exactly the same as the prediction from the unconstrained RS theory [52, 13]. In contrast, the exact capacity of Ising perceptron with replica-symmetry breaking is $\alpha_c \approx 0.83$. For comparison, we have included these values in Fig.12(a), as well as the capacity found by the state-of-the-art supervised learning algorithm (Stochastic Belief Propagation, SBPI [10]) for Ising perceptron.

### A.5.2 Sparse distributions

For a teacher with sparse distribution, $p(w_t) = (1 - \rho)\delta(w_t) + \frac{\rho}{\sqrt{2\pi}\sigma_t w_t} \exp\left\{ -\frac{(\ln w_t - \mu_t)^2}{2\sigma_t^2} \right\}$. We found that the simulations start to deviate from the theory, and the reason might be due to replica symmetry breaking. In Fig.13, we use the optimal prior learning paradigm similar to main text Fig.5. We see that our RS theory no longer gives accurate prediction of overlap in this case.