# OpenReview forum: "A theory of weight distribution-constrained learning"
_NeurIPS.cc/2022/Conference — NeurIPS 2022 Accept_

### Official Review · Reviewer_hTnp · 2022-07-07

**Rating:** 7
**Confidence:** 2
**Soundness:** 4 excellent
**Presentation:** 3 good
**Contribution:** 3 good

**Summary:**

The authors compare the capacity of a perceptron with normally distributed weights to the capacity of a perceptron with weights distributed according to a specified distribution. They show analytically that the difference in capacity is a function of the Wasserstein distance between the normal distribution and the specified distribution. Based on their analysis, the authors derive an algorithm for optimizing the weights of a perceptron subject to a specified weight distribution, which they analyze using tools from optimal transport theory. The authors consider a perceptron whose positive/negative weights are constrained to have log-normal distributions and show that the optimal parameters of the log-normal distributions are close to experimentally observed parameters from connectomics data. Finally, they apply their theory to a student-teacher network and compute the optimal prior on the student network weights given knowledge of the distribution of the teacher network weights.

**Questions:**

Given the discrepancy between experimentally observed weight distributions and the theoretically optimal distribution for the perceptron, why is the natural problem (from the perspective of a computational neuroscientist) to understand the performance of the perceptron under a constrained weight distribution?

Does your learning algorithm suggest how a biological circuit would enforce the distribution constraint (specifically, the optimal transport map step)?

Is there prior work along these lines?

Minor questions:

Why are Eq (4) and Eq (5) equivalent? Because the moments of the distributions are fixed to 1? I think this could be clarified.

In Eq (7) and Algorithm 1, what is $\mu$ summed over? The current batch? It current reads more like GD than SGD.

**Limitations:**

I do not foresee any direct negative societal impact of their work.

**Strengths And Weaknesses:**

As far as I can tell, this work is original (though this is not my specific area of expertise) and their technical arguments are clear (though I did not read the appendices carefully).

My main concern is with the motivation of the paper. In the second paragraph of the introduction, the authors first cite work stating that near capacity, an unconstrained perceptron classifying random input-output associations has normally distributed weights. Then they cite work showing that connectomics data indicates that weight distributions are not normally distributed, but rather closer to log-normally distributed. From my point of view, the natural conclusion from these theoretical and experimental works is that the perceptron is the wrong model with which to understand the log-normal weight distribution. A potential question is then: "What is the simplest task that is optimized when the weights are log-normal?" However, the authors come to an alternative conclusion: "The discrepancy between theory and current experimental result... call for a theory of perceptron learning that incorporates the knowledge of global structural information." Why is this the natural conclusion? Are the authors assuming that the log-normal distribution of the weights is a constraint that is not due to the learning task? I think this needs to be clarified.

This being said, I believe the results are interesting and worthwhile from a theoretical perspective, but the connections to brain function are less clear.

---

> ### Author Response · Authors · 2022-08-02
> **Response to Reviewer hTnp**
>
> > “My main concern is with the motivation of the paper… I think this needs to be clarified”
>
> Thanks for bringing up these motivational questions, please refer to overall comments (1)-(4).
>
> > “Given the discrepancy between experimentally observed weight distributions and the theoretically optimal distribution for the perceptron, why is the natural problem (from the perspective of a computational neuroscientist) to understand the performance of the perceptron under a constrained weight distribution?”
>
> There have been various attempts in the literature to explain the origin of this observed lognormal distribution, and here we do not attempt to add another one. Rather, we take this observed distribution as a feature and try to incorporate it into a well-established model, perceptron, and by doing so hopes to make the perceptron more biologically realistic. Please also see overall comments (1).
>
> > “Does your learning algorithm suggest how a biological circuit would enforce the distribution constraint (specifically, the optimal transport map step)?”
>
> Thank you for pointing out this important limitation. Unfortunately, the answer is no, we have not suggested any biological mechanism that can enforce the distribution constraint.
>
> > “Is there prior work along these lines?”
>
> There has been prior work on incorporating sign-constraints into perceptron learning, notably [Amit et al. (1989)] and [Brunel et al. (2004)], and discussions on what functional principles one can learn from synaptic weight distribution in biological neural networks [Barbour et al. (2007)]. However, to our best knowledge we are not aware of prior work that incorporate distribution constraints into perceptron learning.
>
> > “Why are Eq (4) and Eq (5) equivalent? Because the moments of the distributions are fixed to 1? I think this could be clarified.”
>
> Yes, thanks for pointing this out. Eq (4) and (5) are equivalent because the second moments are fixed to be 1, we gave detailed derivations at the end of Supplementary Materials A1.1 (just above A1.2).
>
> > “In Eq (7) and Algorithm 1, what is μ summed over? The current batch? It current reads more like GD than SGD.”
>
> In Eq (7), μ sums over from 1 to number of examples P, in this simplest setting it is GD rather than SGD. However, in practice we can also implement SGD by using mini-batches, see Supplementary Materials A4.
>
> **References:**
>
> [1] Amit, D. J., Wong, K. M., & Campbell, C. (1989). Perceptron learning with sign-constrained weights. Journal of Physics A: Mathematical and General, 22(12), 2039.
>
> [2] Brunel, N., Hakim, V., Isope, P., Nadal, J. P., & Barbour, B. (2004). Optimal information storage and the distribution of synaptic weights: perceptron versus Purkinje cell. Neuron, 43(5), 745-757.
>
> [3] Barbour, B., Brunel, N., Hakim, V., & Nadal, J. P. (2007). What can we learn from synaptic weight distributions?. TRENDS in Neurosciences, 30(12), 622-629.

---

> > ### Comment · Reviewer_hTnp · 2022-08-04
> > **Response to rebuttal**
> >
> > Thank you for your response and clarifications. I have another comment/question:
> >
> > - Your algorithm for optimizing the network while maintaining a specific weight distribution seems relevant for testing other properties of the network (e.g., adversarial robustness). For example, constraining neural representations to have a power law covariance spectrum improves robustness of the network to adversarial examples [1]. Have you thought about testing your algorithm on such tasks (in addition to generalization in the teacher-student paradigm)?
> >
> > - Minor: Line 276: "parameterized" is misspelled.
> >
> > [1] Nassar, Josue, et al. "On 1/n neural representation and robustness." Advances in Neural Information Processing Systems 33 (2020): 6211-6222.

---

> > > ### Author Response · Authors · 2022-08-08
> > > **We thank the reviewer for the intersting suggestions, which should be explored in future extension of the present work.**
> > >
> > > Thanks for the interesting suggestions.
> > > 1. Robustness-in our setting robustness to noise is determined largely by the margin which is incorporated in our theory. Specifically, our theory predicts that Gaussian distribution of weights confers the maximal margin (for a given load). However, it would be interesting to explore whether there are types of adversarial perturbations for which non Gaussian weights might be superior.
> > > 2. Covariance spectrum-thanks for pointing out the interesting reference. A possible analog in our setting is to constrain the distribution of pre-activations of the output. The 'default' theory predicts mixture of truncated Gaussians (as shown in our work). It would be interesting to explore the consequences of constraining the distribution of these fields away from this default.

---

> > > > ### Comment · Reviewer_hTnp · 2022-08-08
> > > > **Thanks for your response, increasing my score by 1 point**
> > > >
> > > > I agree this is best left for future work.
> > > >
> > > > After more thought, I believe the results in this work are ultimately of interest to the theoretical neuroscience community, so I have increased my score by 1 point.

---

### Official Review · Reviewer_8wH5 · 2022-07-11

**Rating:** 8
**Confidence:** 3
**Soundness:** 4 excellent
**Presentation:** 3 good
**Contribution:** 4 excellent

**Summary:**

The paper develops a theoretical framework for weight distribution-constrained perceptron learning using replica method. They compute capacity based on the proposed distribution, and develop a modification to SGD to satisfy the weight distribution constraints. Their theory is verified in simulations, and it allows to compute the optimal weight prior in the student-teacher learning case. Moreover, its predictions match the experimentally observed weight distributions in mouse primary auditory cortex.

**Questions:**

### General suggestions
Sec. 5: it'd be nice to see a discussion on why you're comparing the auditory cortex to a perceptron in the first place. Does the data from [27] come from a specific subarea known to perform just classification? If it's the whole auditory cortex, why do you think it's comparable?

Lines 99-100: worth noting that the number of synapses per neuron is at most 1k-10k, depending on the area. The large N limit with many memories presumably makes the number of synapses infinite too. (It's a small point that doesn't invalidate any of the results; it's just worth keeping in mind when doing capacity calculations, in my opinion.)

Sign constrained in Fig. 5a: could you fix the drop of the blue line by reducing the initial learning rate?

### Small corrections

Table 1 is really a figure…

Line 121: reiterate that it holds only when the variance of w is one. Otherwise, it’s confusing.

Line 233: put parenthesis for arccos

Line 241: capitalize Gaussian

Appendix placeholders should be removed from the main pdf

**Appendix**

Figs. 5-6: separation, not seperation in the titles

Eq. 2: $\lambda$ should be $\rho$

Eq. 3: better switch index i to something else (you already have the imaginary unit)


**Limitations:**

The authors adequately addressed the limitations and potential negative societal impact.

**Strengths And Weaknesses:**

### Strengths
1. Capacity calculations are rigorous and in good agreement with simulations.
2. Student-teacher results on optimal priors are interesting and non-trivial.
3. The match with mouse data for E/I networks is really interesting, although it's not clear to me why the auditory cortex should look like a big perceptron.

### Weaknesses
1. The results of this paper might be hard really to extend to multi-layer network (which is not surprising)
2. ...and also hard/tedious to extend to other distributions, as it would require a replica calculation (although some ground work for that is done in the appendix)

These are not crucial problems, however.

### Summary
I think it's an interesting paper with a solid theoretical results, good experiments and an interesting connection to real neural data; 8 (strong accept).

---

> ### Author Response · Authors · 2022-08-02
> **Response to Reviewer 8wH5**
>
> > “1. The results of this paper might be hard really to extend to multi-layer network (which is not surprising)”
>
> We can straightforwardly apply the constraints to multiple neurons within the same layer based on results from this paper. However, indeed it is the case that applying our framework to multi-layer network requires more effort, a direction that we are currently pursuing.
>
> > “2. ...and also hard/tedious to extend to other distributions, as it would require a replica calculation (although some ground work for that is done in the appendix)”
>
> Actually, our analytical theory and the replica calculation is **for general distributions**. Plugging in a new distribution Q does not require one to redo the replica calculation, and one only need to write down the equation for cumulative distributions Q(w) = P(x) and solve for w(x), where x is a Gaussian variable and P(x) is the Gaussian CDF. In this sense, all the hard work has been done and one can readily apply our theory and algorithm to any distribution of interest.
>
> > “Sec. 5: it'd be nice to see a discussion on why you're comparing the auditory cortex to a perceptron in the first place. Does the data from [27] come from a specific subarea known to perform just classification? If it's the whole auditory cortex, why do you think it's comparable?”
>
> We chose mouse auditory cortex data because it provided detailed parameter values of the E and I distributions. Furthermore, as we explained in overall comment (1), sensory cortices may multiplex memory functions with pure feedforward sensory filtering. This is particularly true in auditory cortex which is known to be a relatively high stage in the auditory processing hierarchy as argued above. Here we use capacity as a proxy to understand the information storage capabilities of neural networks, also see more in overall comments (3).
>
> > “Lines 99-100: worth noting that the number of synapses per neuron is at most 1k-10k, depending on the area. The large N limit with many memories presumably makes the number of synapses infinite too. (It's a small point that doesn't invalidate any of the results; it's just worth keeping in mind when doing capacity calculations, in my opinion.)”
>
> Thanks for pointing out this fact and bringing this important point to attention. Indeed, our theory assumes large N limit. However, as we show in Supplementary Materials A5.1, N=500 is already a reasonably good approximation to the large N limit, with small finite size effects, and our theory is in good agreement with the numerical simulation. 1k-10k lies well in the regime that the large N limit is valid and can be described by our theory.
>
> > “Sign constrained in Fig. 5a: could you fix the drop of the blue line by reducing the initial learning rate?”
>
> Thanks for the suggestion, but unfortunately the drop cannot be fixed by choosing a small initial learning rate. Fig.5(a) shows the final (converged) solution, not intermediate solutions, and the nature of the drop is fundamentally due to the fact that max-margin is a bad regularizer in the data-limited (small alpha) regime, small initial learning rate does not help to avoid converging to this final (bad) max-margin solution.
>
> And thank you for all the corrections and noticing the typos, we have corrected in the updated version.

---

> > ### Comment · Reviewer_8wH5 · 2022-08-06
> > **Response to the authors**
> >
> > Thank you for your response! I'm happy with the clarifications here and in other comments, so I'm leaving the same score (8).

---

### Official Review · Reviewer_X5tG · 2022-07-11

**Rating:** 4
**Confidence:** 4
**Soundness:** 3 good
**Presentation:** 2 fair
**Contribution:** 2 fair

**Summary:**

The manuscript presents a generalization of traditional perceptron analysis, under constraints over the marginal distribution of weights, showing that perceptron capacity is negatively affected, the more so as the target weight distribution deviated from the unit normal.  Furthermore, a SGD-type algorithm is provided for the solution

**Questions:**

- what does this have to do with biological weight statistics: do you view connectomics data as a source of additional constraints for learning? and if so how does one resolve the apples vs oranges issue of the rest of the statistics being mismatched (see above)?
- i did not understand what exactly do the 'experimental measurements' correspond to in fig4a. please clarify. what does the notion of perceptron capacity is supposed to correspond to in mouse auditory cortex?
- what brain process does the student teacher (a perceptron version of mimic task) setup correspond to?
- isn't it trivially expected that an accurate prior (true weight distribution) should improve performance in this task?
- how is the 'optimal prior' defined exactly?
- is there any relevance of these constraints for machine learning practice?
- is there any benefit of log-normality of weight distributions in this framework?

**Limitations:**

No issues

**Strengths And Weaknesses:**

Strengths:
- interesting capacity analysis in that it combines classic tools (Gardner) with optimal transport
- optimization algorithm that goes with it: standard logistic regression combined with steps ensuring marginal statistics match the target distribution(s) as close as possible

Weaknesses:
- although the math seems sound enough, the link to biology remains fuzzy throughout, at even the broadest level of why is perceptron capacity a useful measure for anything biological, what the student teacher setup is supposed to correspond to, or most importantly the reasons for nonnormality in biological weight distributions
- i found  the arguments on the discrepancy between the marginal statistics of weight strengths in perceptrons trained on a family of random classification tasks and biological weight distributions and the need to unify the two to be fundamentally suspicious; given the discrepancy in computational tasks (input statistics, nature of ecologically relevant tasks, etc), all the heterogeneity in biological units that is in no way modeled in the theory, plus constraints on the learning processes themselves it is not clear at all what weight constrained perceptrons have to do with the data. Also the directionality of the argument seems to flip from the marginal weight distributions being a pre-defined constraint or an explanandum. More generally, the introduction was poorly written at if left unclear what the point of the exercise is exactly.
- accessibility: significant parts of the results text require the reader to be quite familiar with statistical mechanics in general and the results of Gardner on perceptron capacity in particular; this limits somewhat readability for what i'd consider to be the typical neurips reader. Variables are not always clearly defined, which does not help either (e.g. line 104 'k').
-  main text figures were rather uninformative, the numerical vs analytical key comparisons were relegated to the appendix.
- section 5.2 is very difficult to follow, with not enough information provided to be able to judge the reasonableness of the claims

---

> ### Author Response · Authors · 2022-08-02
> **Response to Reviewer X5tG**
>
> > (i) “although the math ... biological weight distributions”;
> > (ii) “what does this have to do with ... mismatched (see above)?”
> > (iii) “i found the arguments ... exercise is exactly.”
>
> Thank you for raising these questions, please see the overall comments (1), (3), and (4). We are not trying to explain the origin of the observed lognormal distribution in biological neural networks, but rather try to understand what this empirical fact of synapses in the brain can tell us about computational properties of neurons in the context of perceptron, and also what we can learn from it by linking structure constraints to neural network functions such as classification capacity.
>
> > “accessibility: significant parts of the results text require the reader to be quite familiar with statistical mechanics in general and the results of Gardner on perceptron capacity in particular; this limits somewhat readability for what i'd consider to be the typical neurips reader. Variables are not always clearly defined, which does not help either (e.g. line 104 'k').”
>
> Thank you for pointing out these issues. We had to delegate a lot of background knowledge on the statistical mechanical theory to the supplementary materials due to the page limit. As for the specific variable in line 104 'k', it refers to the k-th Fourier mode introduced earlier in the same line when we perform the Fourier transform. We have clarified this in the revised text.
>
> > “main text figures were rather uninformative, the numerical vs analytical key comparisons were relegated to the appendix.”
>
> We apologize for not being clear enough for the main text figures. We have updated the main text to more accurately describe the comparison between theory and simulation.
>
> > “section 5.2 is very difficult to follow, with not enough information provided to be able to judge the reasonableness of the claims”
>
> We have further expanded Section 5.2 in Supplementary material A3.3.
>
> > “i did not understand what exactly do the 'experimental measurements' correspond to in fig4a. please clarify. what does the notion of perceptron capacity is supposed to correspond to in mouse auditory cortex?”
>
> The experimental measurements in Fig.4(a) refers to the parameters in the lognormal family of distributions that best fit the experimentally measured connectivity data. As we explained in overall comments, lognormal distribution is ubiquitous in the brain. We chose mouse auditory cortex data because it provided detailed parameter values of the E and I distributions. Furthermore, as we explained above, sensory cortices may multiplex memory functions with pure feedforward sensory filtering. This is particularly true in auditory cortex which is known to be a relatively high stage in the auditory processing hierarchy and exhibits abundant contextual modulation, see e.g., [12].
>
> > “what brain process does the student teacher (a perceptron version of mimic task) setup correspond to?”
>
> Please refer to overall comments (4). We aim to understand the generalization performance of the network under structural constraints and try to answer the question of how faithful one can infer the network connectivity given input-output relations. As mentioned above, teacher-student architectures have been a very fruitful toy model of generalization in neural networks.
>
> > “isn't it trivially expected that an accurate prior (true weight distribution) should improve performance in this task?”, “how is the 'optimal prior' defined exactly?”
>
> The answer is (somewhat counterintuitively) no. We show in Fig.5(b)-(c) that the best prior to use is not always the true weight distribution. The `optimal prior’, defined to be the prior within the family of distributions that we consider that gives the lowest generalization error (or equivalently the highest overlap), changes with the amount of data ($\alpha$), and only approaches the true weight distribution (of the underlying target rule) at the limit of infinite amount of data. Therefore, this statement is not trivially expected, and especially not the case when data is limited.
>
> > “is there any relevance of these constraints for machine learning practice?”
>
> We believe that there is. These constraints are applicable to artificial neural networks with restrictions on the range and distribution of synaptic weights due to software or hardware constraints. Also, in the study of generalization performance, when one has a strong prior on the ground-truth data distribution (for example, prior knowledge about the rule that generates the data), incorporating distribution constraints would improve generalization performance.
>
> > “is there any benefit of log-normality of weight distributions in this framework?”
>
> Please refer to overall comments (1).

---

> > ### Comment · Reviewer_X5tG · 2022-08-06
> > **Post rebuttal comments**
> >
> > Thank you for the clarifications, unfortunately while all questions have been addressed the answers reinforce the discrepancy between the mathematical formalism and the biological system being analyzed. Moreover the relevance of the analysis outside the original scope remains unclear. I have adjusted my score slightly to reflect these persisting issues.

---

> > > ### Author Response · Authors · 2022-08-08
> > > **Our work's assumptions follow the established line of theoretical neuroscience research. It is relevan to biological as well as to artificial nets.**
> > >
> > > The paper is not about log-normal distribution and is exclusively about biology. It is about learning with priors on weight distribution, a key point which seems to have been overlooked. Part of the motivation is the observed log-normal distribution observed in biology. Another motivation addressed in the paper is learning a linear classification rule.. It simply says the network learns a classification rule which itself is based on linear weighing of the inputs with unknown weights. The task of the network is in a way to discover the unknown 'target' weighing of the inputs. There is no reason why networks in the brain do not encounter such tasks. Other applications are cases where the weights are bounded in their value- which have nothing to do with log normal. Such and other constraints are relevant also to artificial neural networks.
> > > Log normal in biology-having non-random inputs does not necessarily imply log normal weights. For instance, depending on the details of the learning, the weight distribution of artificial DCNNs that learn to classify objects from highly structured ImageNet images is close to Gaussian! Thus, our hypothesis that in biology log normal distribution comes from structural constraints rather than learning  is not implausible although as stated above not essential to our work. The ubiquity of this distribution is a suggestive indication.
> > > In summary, despite the abundant citations of previous neuroscience work on similar simplifying assumptions, oddly enough the reviewer claims it reinforces the discrepancy... with no explanation. It would have been more respectful to address our points.  It is hard to respond by attempt at mind reading.

---

### Official Review · Reviewer_42gB · 2022-07-15

**Rating:** 4
**Confidence:** 2
**Soundness:** 2 fair
**Presentation:** 3 good
**Contribution:** 2 fair

**Summary:**

In this paper, the authors:
- provide theory for memory capacity (at a given classification margin) for perceptrons with distribution-constrained weights, and that capacity loss from optimum can be phrased as the Wasserstein distance between the target distribution and standard normal
- design a novel learning algorithm that account for both classification loss and distribution matching (via optimal transport)
- apply to a perceptron with balanced E & I input populations
- investigate the optimal prior for a student neuron in a task where the student neuron is asked to match the teacher's weight distribution given only the teacher's input-output response

**Questions:**

As stated above, the question of how task optimization and weight distribution matching can be simultaneously achieved, given experimental observations of log-normally distributed weights, is a very important question. However, the paper starts with this motivation, but essentially solves the problem for a single perceptron (or, uni-dimensional output logistic regression) with random inputs. I understand that assumptions are necessary for such theoretical work, which I believe is of high quality. But the massive reduction in scope feels almost like a bait and switch (though I'm sure the authors do not intend for this to be the case). At the very least, some investigation on a system that relaxes one or several of the assumptions should be done, even if the theory does not end up so clean:

- would the theory / simulation results hold for non-random inputs? Basically in no scenario does the real brain have to perform classification on perceptually random inputs, so at the minimum, a toy experiment on structured inputs would be justified (MNIST 0 vs 1, for example).

- similarly, the paper is motivated by neural "networks", so how do the results hold up for more than a single neuron? For higher dimensional outputs? Do compensation of weights for different neurons violate the theory?

- I don't see how the results of Figure 5 is relevant to the overall story. Perhaps more justification or clarity in writing needs to be provided

- in the learning algorithm, why does distribution matching have to occur separately from weight optimization? I suppose it fits nicely with the optimal transport theory, but simply optimizing a loss function that combines task performance and, e.g., DKL or Wasserstein distance between the weight distributions, simultaneously would be naively reasonable, and no less biologically implausible (since individual weights do not know of the other weights and certainly does not actively converge to some target lognormal distribution).

**Limitations:**

Some discussion of future works, but very little discussion of limitations in section 6. No discussion of societal impact (but none necessary)

**Strengths And Weaknesses:**

originality: in my limited knowledge, I do not know of other related works that investigate distribution-constrained weights of a network from a theoretical / statistical mechanics point of view

quality: the proof itself (section 2.2) is beyond me, but at a superficial level, the main result (Fig 2-4) seem solid and that simulations support the theoretical prediction. I don't really get how the teacher-student learning paradigm is relevant, however. It seems to be a bit of a contrived setting: even though it makes sense from a decoding perspective (as a neuroscientist trying to infer the weights of a neuron given only its input and outputs), no biological neuron would be required to do this as the student.

clarity: the work is clearly presented, with high quality and thoughtful visualizations, as well as an easy-to-follow high level narrative.

significance: the high level motivation of distribution-matched learning is clearly stated, and of significant interest to the neuroscience community at large. However, the amount of assumptions / simplifications required to arrive at the actual results dampens my enthusiasm (expanded below)

---

> ### Author Response · Authors · 2022-08-02
> **Response to Reviewer 42gB**
>
> > “would the theory / simulation results hold for non-random inputs? Basically in no scenario does the real brain have to perform classification on perceptually random inputs, so at the minimum, a toy experiment on structured inputs would be justified (MNIST 0 vs 1, for example).”
>
> We expect the qualitative conclusions we arrived at, namely that the capacity reduces as the imposed distribution constraint moves further away from Gaussian in a manner tracked by the Wasserstein Distance, and that generalization performance improves when one incorporates appropriate prior structural information into training, will hold when one considers the case for non-random input.
>
> > “similarly, the paper is motivated by neural "networks", so how do the results hold up for more than a single neuron? For higher dimensional outputs? Do compensation of weights for different neurons violate the theory?”
>
> Please refer to overall comment (1). In addition, we are performing a follow-up study on applying distribution constraints on a population of neurons. The main conclusions we arrive at for single neuron hold for population of neurons. Although our theory for classification tasks assumes the input to be one-dimensional, our DisCo-SGD algorithm can be applied to higher-dimensional outputs. The compensation of weights for different neurons interestingly might (tentatively) lead to heterogeneity in the neural population.
>
> > “I don't see how the results of Figure 5 is relevant to the overall story. Perhaps more justification or clarity in writing needs to be provided”
>
> Fig.5(a) shows the generalization performance of perceptron is improved when incorporating structural constraints, and Fig.5(b)-(c) shows that this is not trivially expected, as the optimal prior actually varies with the number of training examples. Overall, we would like to investigate how structural affects function in neural networks, and we believe generalization performance is an important aspect of it. Also see more justification in overall comment (4).
>
> > “in the learning algorithm, why does distribution matching have to occur separately from weight optimization? I suppose it fits nicely with the optimal transport theory, but simply optimizing a loss function that combines task performance and, e.g., DKL or Wasserstein distance between the weight distributions, simultaneously would be naively reasonable, and no less biologically implausible (since individual weights do not know of the other weights and certainly does not actively converge to some target lognormal distribution).”
>
> Actually, straightforward combining of DKL/Wasserstein distance with the cross-entropy loss is difficult because distribution measures are invariant to permutations of synaptic identities, and it is difficult to write down an expression that correctly assigns gradient updates to individual synapses. Therefore, a method that specifies an ordering while also keeps the distribution information is needed, and we choose to adopt the theory of optimal transport.  Nevertheless, we do not claim that this is the only or the best algorithm to learn the task.

---

> > ### Comment · Reviewer_42gB · 2022-08-08
> > **thanks for the response**
> >
> > thanks for responding to my questions:
> >
> > Overall, it seems like there is much work in progress, and while this paper tackles an important problem, and that its theoretical results are of high quality for the single perceptron, I believe the disconnect between the stated motivation (realistic neural networks) and the actual study is too big.
> >
> > - re: the teacher-student setup: I'm not familiar with this literature, but is it really a plausible scenario that a student network (in the real brain) would ever need to learn the weights of a teacher network based on the latter's input-output relationship?
> >
> > - re: invariance to synaptic identities: why is this an important requirement? the overwhelming majority of works ignore synaptic identities during the learning process, and while this is probably not entirely true in the real brain, it seems to be an overkill to require that the distribution shift satisfies optimal transport. In other words, can the authors motivate more clearly why specifying the ordering is important in the biological context?
> >
> > at the high level, it seems to me like the motivation of the study is a bit backwards? The observation is that synaptic weights are distributed log-normally, and we KNOW that the real brain is learning on non-random inputs, but there is no enforcement of the log-normal constraint (how would the brain even do that?) It seems to be a natural consequence of learning, more so than a requirement. Perhaps the authors' intended contribution is simply to study memory capacity given log-normal synapses (through developmental prewiring, e.g.), which comes back to the previous point: why is a learning algorithm that preserves synapse identity important?

---

> > > ### Author Response · Authors · 2022-08-08
> > > **We forcefully argue that our work is applicable to many diverse scenarios, in memory and generalization, in biological as well as artificial nets.**
> > >
> > > 1. The paper is not about log-normal distribution. It is about learning with priors on weight distribution, a key point which seems to have been overlooked. Part of the motivation is the observed log-normal distribution observed in biology. Another motivation addressed in the paper is learning a linear classification rule. The terminology student - teacher may have been confusing. It simply says the network learns a classification rule which itself is based on linear weighing of the inputs with unknown weights. The task of the network is in a way to discover the unknown 'target' weighing of the inputs. There is no reason why networks in the brain do not encounter such tasks.
> > > Other applications are cases where the weights are bounded in their value- which have nolthing to do with log normal. Such and other constraints are relevant also to artificial neural networks.
> > > 2. Log normal in biology-despite the claim of the reviewer, having non-random inputs does not necessarily imply log normal weights. For instance, depending on the details of the learning, the weight distribution of artificial DCNNs that learn to classify objects from highly structured ImageNet images is close to Gaussian!  Thus, our hypothesis that in biology log normal distribution comes from structural constraints is not implausible although as stated above not essential to our work. The ubiquity of this distribution is a suggestive . indication. How does the brain generate it? for instance the brain may maximize the entropy of the log of the weights.
> > > In summary both points 1-2 forcefully argue that our work has applicable to many diverse scenarios.
> > > 3. Invariance of the weight identity. We simply state the obvious-that constraint on distribution is blind to weight identity.
> > > 4. Optimal transport-As stated in our previous response, learning via optimal transport is a nice feature of our algorithm but we do not claim it is the only learning algorithm of the task. We do not think the elegance of an algorithm is necessarily  a bug.

---

### Author Response · Authors · 2022-08-02
**Overall Response**

Thank you so much for all the careful reviews. We noticed a few commonly raised questions by the reviewers, and we will address them here first.

## Overall comments:
**(1)  The motivation of using perceptron model and its biological relevance.**

**1 . Perceptron.**
Perceptron is arguably the simplest model of computation by single neuron and is the fundamental building block for many modern neural networks. Despite the drastic oversimplification, studying the computational properties of (binary and analog) perceptron has been used extensively in computational neuroscience since its dawn, particularly in the cerebellum (as a model of sensory-motor association) but also in cerebral cortex (for generic associative memory functions) (e.g., [1]-[6]).
Briefly, the rationale is that forming associations is considered an ‘atomic’ building block for generic cortical functions.

**2. Log-normal distribution.**
Why lognormal distribution of synaptic strengths is so ubiquitous in biological circuits is an important question. Possible reasons range from biological structural/developmental **constraints** to benefits for certain yet to be elaborated biological computations. Whatever the reason might be, cortical circuits are known to be multiplexing computational tasks, so that it is reasonable to study memory capacity not only in circuits dedicated to memory but also in sensory cortical circuits, as an important computational feature beyond and above the ongoing sensory – motor processing in the same circuit. Importantly, from the perspective of associative memory, non-Gaussian weight distribution is a constraint rather than a benefit and the goal of the paper is **to present for the first time** a quantitative and qualitative theory of the reduction in memory performance due to non-Gaussian weight distribution.
We would like to emphasize that our theory is general and is **not limited to log-normal distributions**. It is thus applicable also to artificial neural networks where due to software or hardware constraints may restrict the range and distribution of synaptic weights. One surprising outcome of the work is that learning with bimodal weight distributions seems to converge properly although extreme bimodal -binary or close to binary are hard to learn even in the perceptron. Also, our theory is not limited to the single neuron model – the perceptron. As mentioned above, it applies to memory capacity in RNNs. A version of our theory as well as the DisCo-SGD algorithm that incorporates large populations of neurons with heterogeneous connectivity is an ongoing extension of this work.


**(2) The motivation of using random input data.**

We make these simplifying assumptions in order to study theoretically the connection between structural constraints and neural function. The tasks we consider here (capacity and generalization) have been extensively studied in the past assuming similarly simplifying assumptions about the data. Our goal is that by investigating an analytically tractable model we can obtain insights that can be of relevance in more realistic settings. As a first step away from the simplest data assumption made in the main text, we considered the case of biased input (although still random) and sparse label in supplementary materials A1.3. Previous work on brain related perceptron modeling (cited above and more) assumes unstructured data.
Indeed, in biological memory systems, the heavily correlated sensory data is undergoing heavy preprocessing including massive decorrelations.


**(3) The motivation of studying capacity.**

Perceptron memory capacity sets a tight bound on the memory capacity in recurrently connected neuronal circuits with application to cortex and hippocampus (e.g., [7]-[9]).
We would like to understand how the information storage property of neurons is affected by the imposed structural constraints, and by observing the optimality in its performance to infer the extent to which actual biological neurons are configured toward maximizing information storage ability.

**(4) The motivation of using teacher-student setup.**

The motivation of introducing the teacher-student setup is two-fold: 1) to study generalization performance of perceptron in classification task; 2) to address a circuit reconstruction task: given experimentally measured stimuli-response relations in artificial or biological networks, how faithfully one can infer the underlying structural connectivity. In the first case, the teacher-student setup is a well-known ‘toy model’ for studying generalization performance (see e.g., [9] and [10]; and more recent extensions to DNNs-e.g., [11]), and we choose to study generalization because it quantifies the quality of learning performed by the perceptron. In the second case, we consider the teacher as a biological neural network with some experimentally measured data while the ground truth circuit is still largely unknown.

---

> ### Author Response · Authors · 2022-08-02
> **References of Overall Response**
>
> [1] Marr D (1969) A theory of cerebellar cortex. J Physiol 202:437–470.
>
> [2] Albus JS (1971) A theory of cerebellar function. Math Biosci 10:25–61
>
> [3] Brunel, N., Hakim, V., Isope, P., Nadal, J. P., & Barbour, B. (2004). Optimal information storage and the distribution of synaptic weights: perceptron versus Purkinje cell. Neuron, 43(5), 745-757.
>
> [4] Chapeton J, Fares T, LaSota D, Stepanyants A (2012) Efficient associative memory stor-
> age in cortical circuits of inhibitory and excitatory neurons. Proc Natl Acad Sci USA
> 109:E3614–E3622.
>
> [5] Brunel N (2016) Is cortical connectivity optimized for storing information? Nat Neu-
> rosci 19:749–755.
>
> [6] Bouvier, Guy, et al. "Cerebellar learning using perturbations." Elife 7 (2018): e31599.
>
> [7] Gardner, E. (1987). Maximum Storage Capacity in Neural Networks. EPL, 4, 481-485.
>
> [8] Rolls, Edmund T., Alessandro Treves, and Edmund T. Rolls. Neural networks and brain function. Vol. 572. Oxford: Oxford university press, 1998.
>
> [9] Rubin, Ran, L. F. Abbott, and Haim Sompolinsky. "Balanced excitation and inhibition are required for high-capacity, noise-robust neuronal selectivity." Proceedings of the National Academy of Sciences 114.44 (2017): E9366-E9375.
>
> [10] Lee, Sebastian, Sebastian Goldt, and Andrew Saxe. "Continual learning in the teacher-student setup: Impact of task similarity." International Conference on Machine Learning. PMLR, 2021.
>
> [11] Matiisen, Tambet, et al. "Teacher–student curriculum learning." IEEE transactions on neural networks and learning systems 31.9 (2019): 3732-3740.
>
> [12] Angeloni C, Geffen MN. Contextual modulation of sound processing in the auditory cortex. Curr Opin Neurobiol. 2018 Apr;49:8-15.

---

### Meta-Review · Area_Chair_XxQo · 2022-08-26

**Recommendation:** Accept
**Confidence:** Less certain

**Metareview:**

Reviewers appreciate the novel weight-distribution contrained algorithm, in spite of reservations about potential impact in comp neuro or ML remain.

**Award:**

No

---

### Decision · Program_Chairs · 2022-09-14

Accept